# *Neurofibromin 1* in mushroom body neurons mediates circadian wake drive through activating cAMP–PKA signaling

Pedro Machado Almeida[1,3], Blanca Lago Solis[1,3], Luca Stickley [1], Alexis Feidler [1,2] & Emi Nagoshi [1✉]

Various behavioral and cognitive states exhibit circadian variations in animals across phyla including *Drosophila melanogaster*, in which only ~0.1% of the brain's neurons contain circadian clocks. Clock neurons transmit the timing information to a plethora of non-clock neurons via poorly understood mechanisms. Here, we address the molecular underpinning of this phenomenon by profiling circadian gene expression in non-clock neurons that constitute the mushroom body, the center of associative learning and sleep regulation. We show that circadian clocks drive rhythmic expression of hundreds of genes in mushroom body neurons, including the *Neurofibromin 1* (*Nf1*) tumor suppressor gene and *Pka-C1*. Circadian clocks also drive calcium rhythms in mushroom body neurons via NF1-cAMP/PKA-C1 signaling, eliciting higher mushroom body activity during the day than at night, thereby promoting daytime wakefulness. These findings reveal the pervasive, non-cell-autonomous circadian regulation of gene expression in the brain and its role in sleep.

---

[1] Department of Genetics and Evolution, Sciences III, University of Geneva, 30 Quai Ernest-Ansermet, Geneva 4, CH-1211, Switzerland. [2] Present address: University of Rochester School of Medicine and Dentistry, Rochester, NY, USA. [3] These authors contributed equally: Pedro Machado Almeida, Blanca Lago Solis. ✉email: Emi.Nagoshi@unige.ch

Numerous cognitive, sensory, behavioral, and emotional states exhibit circadian (near 24 h) rhythms in most animal species[1–5]. In humans, misalignment between day–night cycles and internal circadian rhythms due to modern lifestyle choices, such as shift work and social jetlag, correlates with a greater risk of disturbances in sleep, cognitive functions, and mental health[6,7]. Although the basic mechanisms of circadian clocks and the neurological underpinnings of various brain functions have been extensively studied, how the former regulates the latter remains incompletely understood.

*Drosophila melanogaster* offers an unusual opportunity to study circadian regulation of brain function. The fly brain, consisting of ~150,000 neurons and 15,000 glia[8], is numerically and genetically simpler as well as tractable with a battery of genetic tools, but shares many features with the mammalian central nervous system (CNS)[9]. Flies and mammals also share the design principle and some of the components of the molecular clocks, which consist of interlocked transcriptional feedback loops. In the main feedback loop of the *Drosophila* clock, the heterodimeric transcriptional activator CLOCK/ CYCLE (CLK/CYC) drives the expression of the *period* (*per*) and *timeless* (*tim*) genes. The PER and TIM proteins accumulate and dimerize in the cytoplasm, subsequently translocating into the nucleus and inhibiting CLK/CYC activity. In addition to this negative feedback loop, PDP-1 and VRILLE (VRI) regulate the transcription of *Clk*, thereby forming a stabilizing loop[10]. Furthermore, many features of the network organization and function of the mammalian central pacemaker, the suprachiasmatic nucleus (SCN), show similarities with the fly pacemaker neurons, such as the importance of peptidergic signaling[11] and cAMP[12]. These similarities in molecular and neuronal mechanisms suggest that studies in flies can shed light on the mechanistic principles of the circadian system in mammals[13].

Flies display circadian rhythms in a variety of behaviors and brain functions, such as locomotor activity, learning and memory, sleep−wake cycles, feeding, courtship, and mating[14]; with numerous brain areas carrying out these functions. Importantly, unlike mammals where virtually all cells have clocks capable of cell- and tissue-autonomously generating rhythms, only ~150 circadian pacemaker neurons and 1800 glia within the fly CNS have molecular clocks[15]. Therefore, circadian pacemaker neurons not only control rhythmic locomotor behavior but also presumably input time-of-day information to non-clock cells in different brain areas to modulate their function and output in a circadian fashion. This regulation could occur via direct neuronal connections or indirectly through interneurons or hormonal/ peptidergic signaling.

Recent studies have identified several non-clock neurons that relay the activity of pacemaker neurons to the circuits controlling locomotor rhythms and sleep–wake cycles. One locomotor output pathway involves neurons expressing the neuropeptide, leucokinin (LK)[16]. Another route includes neurons expressing diuretic hormone 44 (DH44) located in the pars intercerebralis (PI)[17]. Additionally, a cluster of dopaminergic neurons, PPM3-EB, was shown to relay the activity of the pacemaker neurons to the premotor center, the ring neurons of the ellipsoid body (EB)[18]. The ring neurons of the EB also play an essential role in sleep homeostasis[19], receiving input indirectly from a subgroup of pacemaker neurons, dorsal neuron 1 (DN1), via a class of tubercular−bulbar (TuBu) neurons[20,21]. Another subclass of pacemaker neuron, lateral posterior neurons (LPNs), is presynaptic to the dorsal fan-shaped body (dFB) of the central complex. LPNs express the allatostatin-A (AstA) neuropeptide as well as glutamate, activating dFB neurons to promote sleep[22]. There are potentially many more brain regions that are under the control of the circadian pacemaker circuit, including the mushroom body (MB).

The MB is a symmetrical structure composed of ~2000 intrinsic neurons called Kenyon cells (KCs), the dendrites of which form a calyx, and their axonal projections converge into a bundle called the peduncle. At the end of the peduncle, KC axons bifurcate to form the vertical (α and α') and horizontal (β, β', and γ) lobes. Three major subtypes of KCs form these lobes: one that projects to the γ lobe and two that have branched axons projecting into the α/β lobes or α'/β' lobes. The MB lobes can be further divided into anatomically and functionally different layers and regions[23–25]. The MB is a major center of information processing in insects, analogous to the cerebral cortex, hippocampus, and piriform cortex in the human brain[26]. Overwhelming evidence establishes the MB as a center for associative learning and memory[27] and regulation of sleep[28–30]. Both of these behavioral processes display daily rhythms[31,32]. A previous study found that the main pacemaker neurons of the circadian circuit, the small ventral lateral neurons (s-LNvs), make physical contacts with some of the KCs[33]; therefore, it is assumed that input from the pacemaker neurons rhythmically modulates MB physiology and their functional output. However, the underlying mechanisms are unknown.

To broadly explore the mechanisms underlying the circadian modulation of MB function, we hypothesized that (1) many, if not all, MB neurons exhibit circadian rhythms in gene expression, despite containing no molecular clocks, and (2) some of these oscillating genes mediate rhythmic function of the MB. The present study tests these hypotheses by conducting circadian RNA-seq analysis of MB neurons, followed by functional analyses of candidate genes with respect to sleep, a process regulated by the MB and occurring with circadian periodicity. We identify a large number of genes expressed with ~24 h period in the MB both under light–dark (LD) cycles and in constant darkness (DD). Gene expression rhythms in DD in the MB are abolished in *period* null (*per⁰*) mutants. *Neurofibromin 1* (*Nf1*), the *Drosophila* ortholog of a tumor suppressor linked to neurofibromatosis type 1 (NF1), and *Pka-C1* encoding the catalytic subunit of cAMP-dependent protein kinase, are among the rhythmically expressed genes in the MB. We further demonstrate that circadian clocks drive cAMP and calcium rhythms in MB neurons, eliciting higher MB activity during the day than at night. Elevated daytime activity of MB neurons is mediated by NF1 and its downstream cAMP–PKA signaling, and in turn promotes wakefulness during the daytime. Sleep is thought to be regulated by the interaction between two processes: the homeostatic sleep drive (Process S) and the circadian process driving wakefulness (Process C)[34]. Our data reveal the pervasive, non-cell-autonomous circadian regulation of gene expression in the brain and highlight its role in the Process C of sleep regulation.

## Results

**Circadian gene expression profiling of MB neurons.** We performed circadian RNA-seq analysis of the MB neurons labelled with GFP using a pan-MB neuron driver, *OK107-GAL4*[35]. Brains were dissected and dissociated every 4 h from flies kept under a 12 h light–dark (LD) cycle and on the second day of constant darkness (DD) following LD-entrainment. On average, 27,000 GFP-positive cells were FACS-sorted at each timepoint, from which poly-A-tailed RNAs were amplified and sequenced following an established protocol[36] (Fig. 1a, Supplemental Data 1). Analysis of the RNA-seq data from biological duplicates using the JTK_CYCLE algorithm[37] identified hundreds of genes expressed rhythmically with a ~24 h period (Fig. 1b, c). A total of 832 genes during LD and 1144 genes during DD were found to be

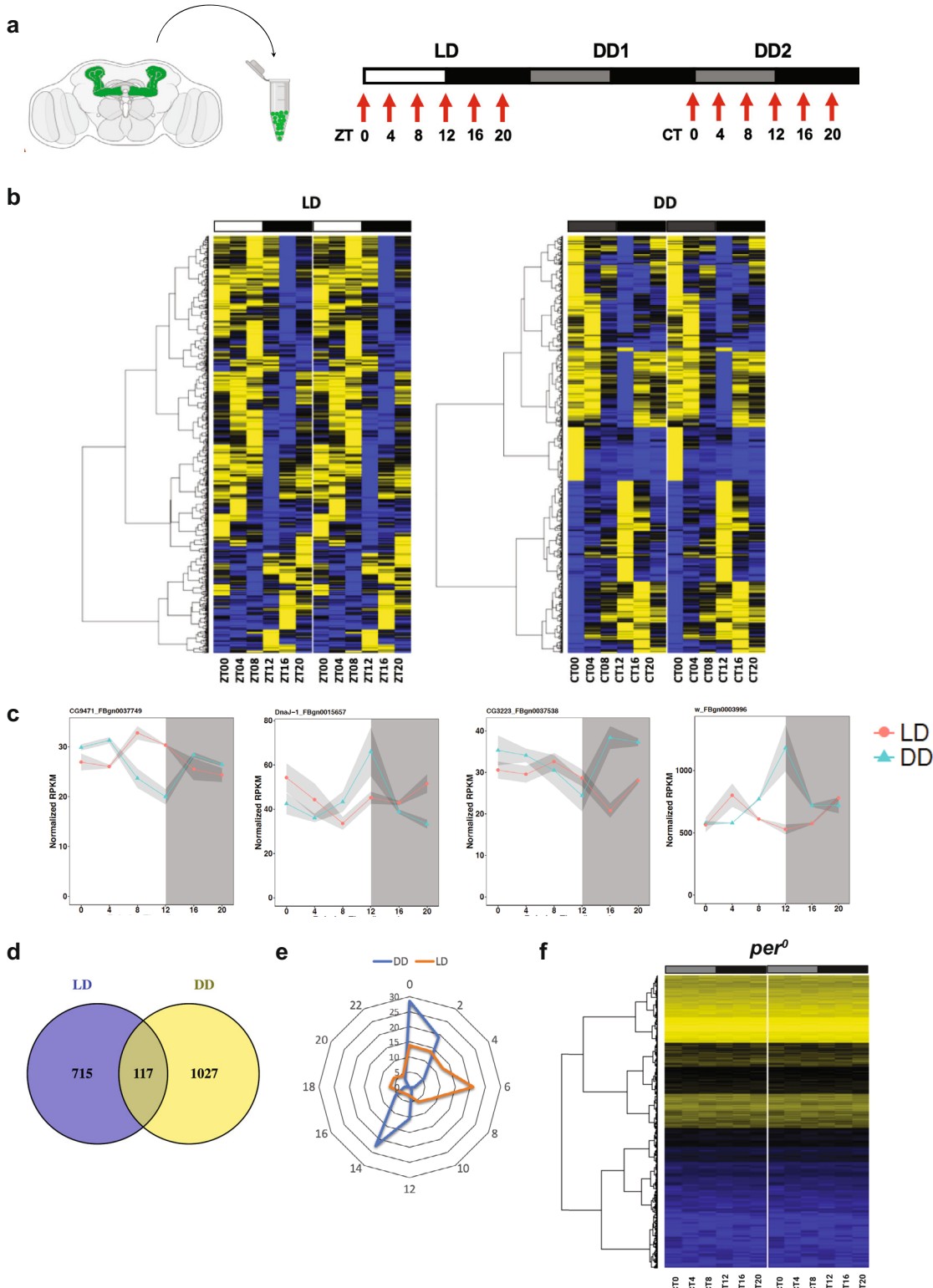

**Fig. 1 Circadian gene expression profiling of MB neurons. a** Schematic representation of the circadian MB neuron cell-sorting RNA-seq assay. **b** Heatmap depicting the genes expressed with circadian oscillations during LD and DD in the MB as identified using the nonparametric JTK_cycle algorithm with a cut-off of *p.adj* < 0.05. Each row represents a gene, and rows are ordered by hierarchical clustering using the ward.D2 method. Yellow and blue colors indicate high and low relative expression levels, respectively. **c** Examples of RNA-seq profiles of cycling genes in LD (red) and DD (blue). *X*-axis indicates ZT or CT. The values are the mean of the normalized RPKM ± SEM. **d** Venn diagram depicting the number of genes expressed rhythmically during LD and DD in the MB. **e** Circular plot representing the peak phases of circadian gene expression in LD (red) and DD (blue). The number on each circle indicates the % of cycling genes peaking at a given ZT or CT. **f** Heatmap representation of genes expressed in DD in MB neurons of *per⁰* flies. Data are analyzed and plotted as in **b**. No gene with a 24 h periodicity was identified using JTK_cycle with a cut-off of *p.adj* < 0.05.

rhythmically expressed, among which 117 genes cycled in both LD and DD (Fig. 1d). These are conservative estimates, given that analysis using the less stringent RAIN[38] and ORIOS[39] algorithms found ~1200 and ~2000 genes, respectively, cycling in both LD and DD.

Importantly, our RNA-seq data show the absence of canonical clock gene expression in MB neurons (Supplemental Data 1). Moreover, RNA-seq analysis of MB neurons in *per0* flies showed no genes expressed with circadian rhythmicity in DD (Fig. 1f and Supplemental Data 2). Therefore, rhythmic gene expression in the MB in DD is non-cell-autonomously controlled by the circadian clocks located elsewhere. The distribution of peak phases was markedly different between the LD and DD conditions: the majority of genes expressed rhythmically in LD (LD cyclers) peaked in the daytime, whereas the phases of genes cycles during DD (DD cyclers) showed a bimodal distribution, peaking around subjective dawn (CT0) or dusk (CT14) (Fig. 1b, e). The data also show that ~85% of LD cyclers became arrhythmic in DD and ~10% of DD cyclers lost oscillations in LD (Fig. 1d), indicating that light–dark cycles can drive gene expression rhythms in the MB and even impede clock-driven mRNA rhythms.

Genes expressed rhythmically in both LD and DD in the MB (dubbed MB LD-DD cyclers) are clock-driven and their rhythms are not masked by LD cycles, hence may represent functionally important genes in the natural environment. We therefore chose to focus on this subset for subsequent studies, starting with a Gene Ontology (GO) term enrichment analysis (Supplementary Table 1). Despite this subset containing only 117 genes, the analysis found enrichment of several GO terms related to the processes known to be regulated by the MB, such as cognition, learning, and memory. Incidentally, these processes are known to undergo circadian modulations[1,2,40]. GO terms related to aminergic neurotransmitter loading and amine transport were also found to be enriched, implicating aminergic transmission in circadian rhythms of MB functions. GO terms in RNA metabolism and transcription initiation were also over-represented, suggesting the transcriptional and RNA-mediated regulation of circadian MB functioning.

**Identification of enriched *cis*-regulatory elements in the MB LD-DD cyclers**. Rhythms of mRNA levels can be controlled by a transcriptional or a post-transcriptional mechanism, or a combination of both. To explore whether either mechanism is involved in cycling gene expression in MB neurons, we first searched for transcription factor binding sites (TFBSs) that are over-represented in the promoter regions of the MB LD-DD cyclers as compared with any other *Drosophila* gene using the oPOSSUM3 tool[41]. A total of 117 TFBSs were found to be enriched in the input dataset, among which 8 exceeded the Z-score threshold (>5.5), 11 exceeded the Fisher-score threshold (>0.55), and only 4 exceeded both; the binding sites of Optix, Twist, Onecut, and Dorsal (Supplementary Table 2). This relatively restricted number of enriched TFBS motifs suggests that many cycling genes could be transcriptionally co-regulated by the same transcription factor.

MicroRNAs regulate the stability and translation of mRNAs and are involved in the functioning of molecular clocks and their outputs in various organisms[42]. At least nine miRNAs are known to regulate *Drosophila* circadian rhythms[42,43], of which six are expressed rhythmically in the brain[42]. Using the microRNA.org database[44,45], we found that all nine circadian-relevant miRNAs have putative targets within the 3'UTRs of the MB LD-DD cyclers (Supplementary Table 3). Thirty two genes out of 117 MB LD-DD cyclers have at least one target site of a circadian-relevant miRNA. Remarkably, *Neurofibromin 1* (*Nf1*), *shaggy* (*sgg*) and

*skywalker* (*sky*) possess target sites for three out of the nine circadian-relevant miRNAs. These bioinformatic predictions suggest that expression or translation of some of the MB LD-DD cyclers may be regulated post-transcriptionally by miRNAs.

**Identification of the MB LD-DD cyclers regulating sleep**. Next, we asked whether any MB LD-DD cyclers play a role in the behavioral processes that show circadian rhythms and are regulated by the MB. We chose to focus on sleep because it is a universal behavioral state regulated by a circadian process, but whose molecular underpinnings remain incompletely understood. To identify genes involved in the regulation of sleep among the MB LD-DD cyclers, we generated a priority list of 21 genes based on criteria including putative functions, expression rhythm strength, and availability of fly stocks (Supplementary Table 4), and knocked them down in the MB using *OK107-GAL4* (UAS-RNAi lines used in this mini-screen are listed in the Method). Sleep of male and female flies was measured over 5 days during LD in the *Drosophila* Activity Monitor (DAM) system and compared with controls that carry the driver only (*OK107/ +*). *w1118* was included in the assay as a reference since most RNAi lines were generated in this background. One cross (*OK107 > CG8735-RNAi*) did not produce viable adult flies and was excluded from this initial screen. Knockdown of five genes in males and six genes in females caused an increase in sleep (Fig. 2a and Supplementary Fig. 1a), but circadian locomotor rhythms remained largely normal, with the exception of *CG8142* (Supplementary Table 5). The genes that showed a sleep phenotype upon knockdown were expressed at different levels and with different temporal patterns in the MB, but all of them increased sleep (Fig. 2a, b). We also selected 12 genes that showed high-amplitude rhythms only in LD and tested their involvement in sleep. None of these 12 LD cyclers affected sleep when knocked down in MB neurons (Supplementary Fig. 1b).

Knockdown of *Pka-C1* and *Nf1* caused an increase in sleep in both males and females (Fig. 2a and Supplementary Fig. 1a). Expression rhythms of *Pka-C1* and *Nf1* mRNAs in the MB are controlled by circadian clocks and peak during the daytime and at the end of the night, respectively, in LD (Fig. 2b, c). Immunostaining of fly brains with antibodies against PKA-C1 and Fasciclin II (Fas II), a marker for MB lobes, revealed the rhythmic accumulation of PKA-C1 protein in the MB lobes in LD and DD in the control (*w1118*) flies and under LD in *per0* (Fig. 3a–c). PKA-C1 rhythms were abolished in *per0* flies in DD (Fig. 3d), consistent with the finding that *Pka-C1* mRNA rhythms are generated by the clocks and light (Fig. 2b, c). In LD, both mRNA and protein levels display bimodal rhythms, and changes in *Pka-C1* mRNA levels precede those of its protein by 4 h (Figs. 2b and 3a). In DD, *Pka-C1* mRNA levels sharply peak at CT0, whereas protein levels rise after CT8 (Figs. 2b and 3b). PKA-C1 protein levels in LD in *per0* show a single peak at ZT8 (Fig. 3c). These results demonstrate the contribution of both circadian clocks and light in shaping the rhythms of *Pka-C1* mRNA and protein, pointing to the possibility that light advances the protein peak and can even create one in the absence of clocks. PKA-C1 was also detected in the pars intercerebralis (PI) in ~50% of flies examined, but it did not cycle in both LD and DD (Supplementary Fig. 2a–c). This observation suggests that, although some cells outside the MB lobes also express *OK107-GAL4*, since non-MB cells account for only a small fraction of *OK107*-positive cells, the RNA-seq data largely reflect gene expression within the MB. We were unable to detect NF1 protein in adult fly brains with several antibodies tested, which is likely due to its low expression levels (see Fig. 2b).

*Pka-C1* encodes a catalytic subunit of cAMP-dependent protein kinase (PKA). PKA activity and sleep are inversely

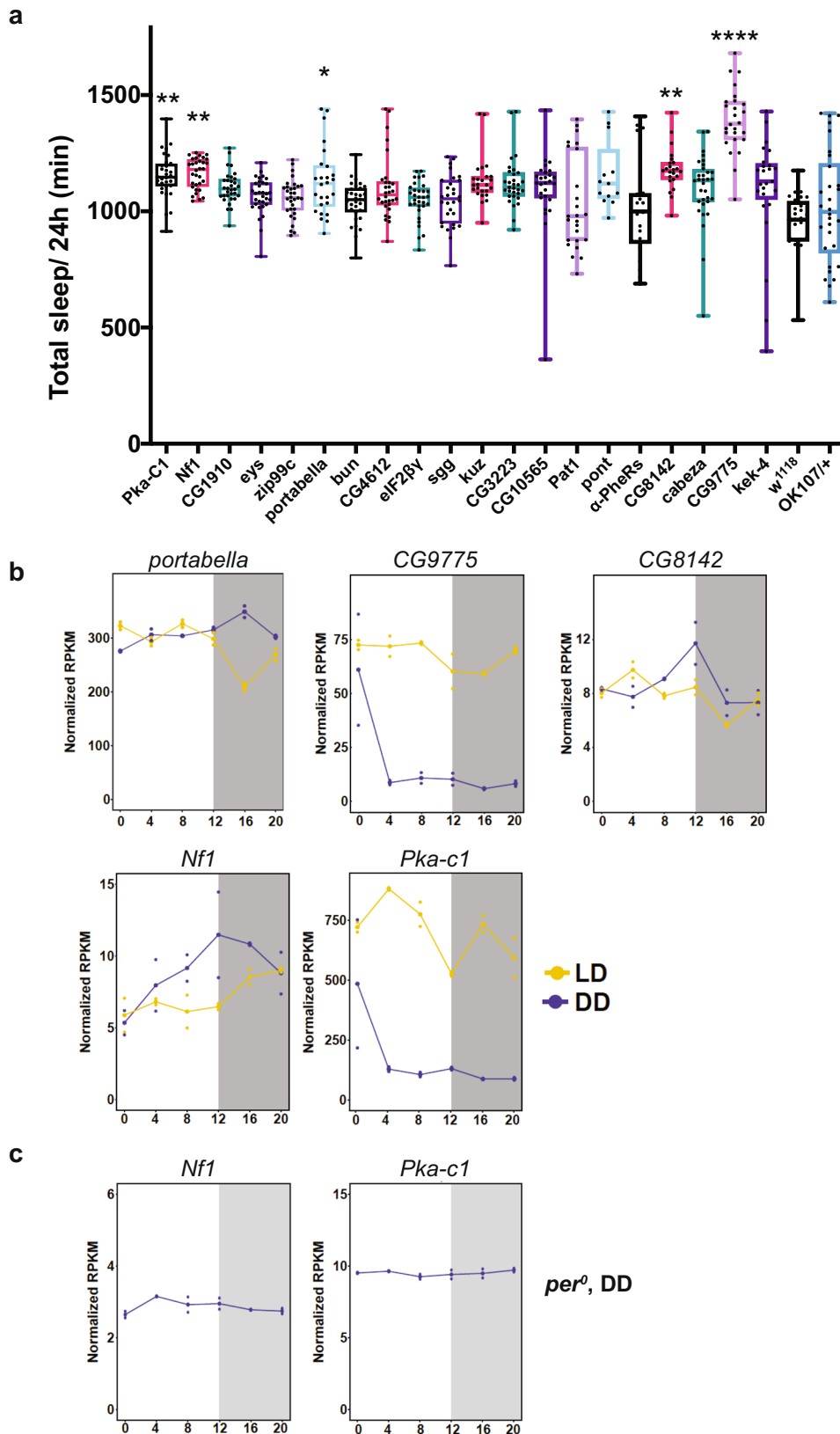

correlated in flies[46], and expression of a constitutively active form of mouse PKA-C1 (Pka-C1 mc*)[47] in the entire MB during adulthood has been shown to decrease sleep[29]. Our findings are consistent with those of previous studies and further define the role of endogenous *Pka-C1* in the MB. *Nf1* is expressed in various cell types and contributes to different neuronal functions, including learning and memory[48], circadian locomotor activity[49–51], grooming, and sleep[52,53]. Curiously, *Nf1* mutations and pan-neuronal knockdown of *Nf1* reduce sleep[50,52,53]. Our data demonstrating that *Nf1* knockdown restricted to the MB is sleep-promoting suggest that NF1 can be sleep-promoting or arousal-promoting depending on the cell type.

**Fig. 2 Screening for MB LD-DD cyclers involved in sleep. a** The total daily sleep of male flies expressing an RNAi transgene with the OK107 driver or $w^{1118}$. The center lines indicate the median, box boundaries are 25th and 75th percentiles, and the whiskers represent the minimum and maximum values. The total amount of sleep per day was compared with $OK107/+$ using the one-way ANOVA with Dunnett's multiple comparisons test (*$p < 0.05$, **$p < 0.01$, ****$p < 0.0001$; $n = 13$–32 flies per group). Source data are provided as a Source Data file. **b** RNA-seq profiles of genes that increase sleep upon knockdown in the MB in males. Each data point represents the normalized RPKM of the indicated gene in one replicate. Connecting lines represent the means of data points. X-axis indicates ZT or CT. RNA-seq profiles in LD and DD are displayed in red and blue, respectively. **c** RNA-seq profiles of Nf1 and Pka-c1 genes in the MB neurons in $per^0$ mutant flies collected during DD2. Data are presented as in **c**.

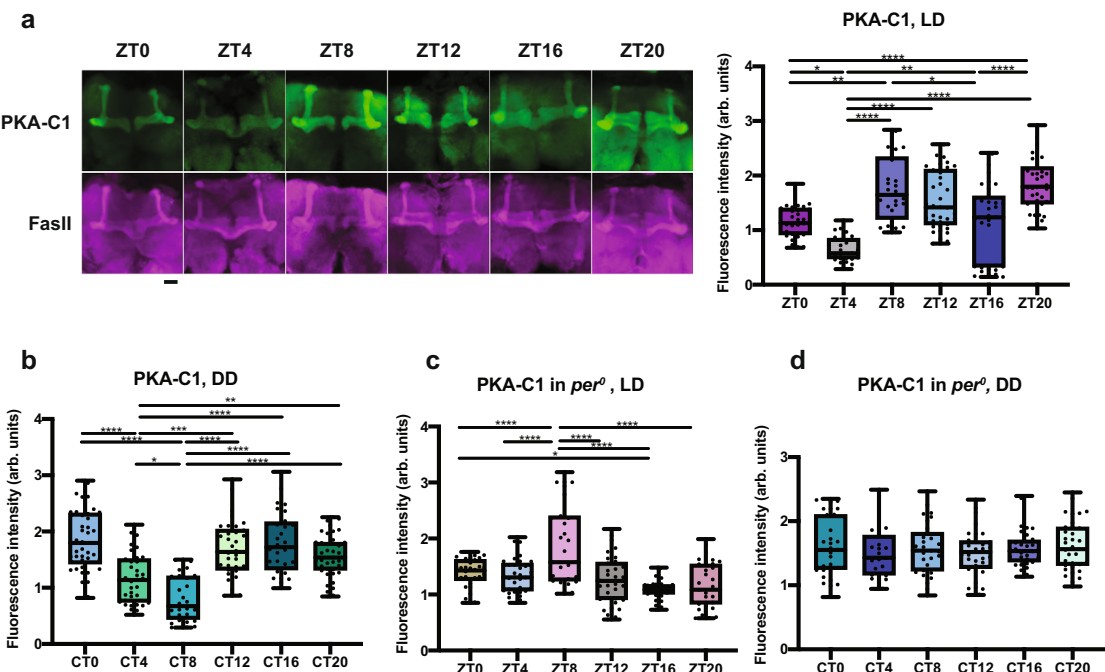

**Fig. 3 PKA-C1 expression rhythms in the MB are regulated by circadian clocks and light.** Staining of fly brains with anti-PKA-C1 and Fas II antibodies demonstrates clock- and light-dependent rhythms of PKA-C1 accumulation in the MB lobes. **a** Left, representative confocal images of the brains of control ($w^{1118}$ background) flies collected at 6 timepoints every 4 h. Green, PKA-C1. Magenta, Fas II. Scale bar, 25 μm. Right, quantification of the PKA-C1 signals in the MB. The center lines of the boxplots indicate the median and the whiskers represent the minimum and maximum values. Dots represent individual values. arb. units, arbitrary units. **b** PKA-C1 levels in the MB in DD. **c** PKA-C1 levels in $per^0$ in LD. **d** PKA-C1 levels in $per^0$ in DD. $n = 30$–40 per group. Based on the distribution of the data, the Kruskal–Wallis nonparametric test followed by Dunn's multiple comparisons test, or the ordinary one-way ANOVA followed by Tukey's multiple comparisons test was used for comparing values between timepoints. *$p < 0.05$, **$p < 0.01$, ***$p < 0.001$, and ****$p < 0.0001$. Source data are provided as a Source Data file.

*Nf1* encodes a large, multidomain protein that is best characterized as a Ras-GTPase-activating protein (Ras-GAP)[54,55]. Additionally, several lines of evidence indicate that NF1 positively regulates cAMP−PKA signaling in various tissues[56–58]. *Nf1* null flies have growth defects, which can be rescued by increasing PKA activity and assimilated by inhibiting the cAMP−PKA pathway[57,59,60], suggesting that PKA acts downstream of, or in a parallel pathway to, NF1. Our initial results that *Pka-C1* or *Nf1* are both rhythmically expressed in the MB and wake-promoting are congruent with the convergent roles of PKA and NF1 and prompt further dissection of their functions in sleep.

**NF1 and PKA-C1 in the MB promote daytime wakefulness**. We repeated the knockdown of *Nf1* and *Pka-C1* in the MB and analyzed different parameters of sleep. As in the initial screen, we drove the expression of the *Nf1* VDRC (KK) RNAi line, which has been successfully used and validated in multiple previous studies[49,52,53], with *OK107-GAL4*. *Nf1* knockdown profoundly increased sleep during the daytime and to a much lesser extent at night (Fig. 4a, b). The increase in daytime sleep was due to a more consolidated sleep, since *Nf1*-knockdown flies had fewer sleep

episodes and increased maximum sleep episode duration as compared with both driver-only and UAS-RNAi-only controls (Fig. 4c–e). Circadian activity patterns were not affected by *Nf1* knockdown in the MB (Supplementary Fig. 3a). Increase in sleep in the *Nf1*-knockdown flies is not due to hypoactivity, as their activity levels during wake were not reduced but rather slightly increased compared with controls (Supplementary Fig. 3b).

*Nf1* mutants display growth deficiency and reduced sleep[50,52,57,60], but it remains unknown whether the growth defect is causally linked to sleep loss. To evaluate whether the sleep-promoting effect of MB-targeted *Nf1* knockdown could be developmental in origin, we restricted knockdown in the MB to adulthood using the drug-inducible MB-GeneSwitch (MB-GS) driver[61]. Knockdown was induced only during adulthood by feeding flies with RU486 after eclosion. The results were similar to those obtained with a constitutive knockdown, showing longer and more consolidated sleep during the daytime (Fig. 5a–f); therefore, NF1 expressed in the MB during adulthood mediates daytime wakefulness.

Knockdown of PKA-C1 in the MB using a second independent RNAi line (*Pka-C1* shRNA) consolidated daytime sleep, as evidenced by the increase in the maximum sleep episode duration

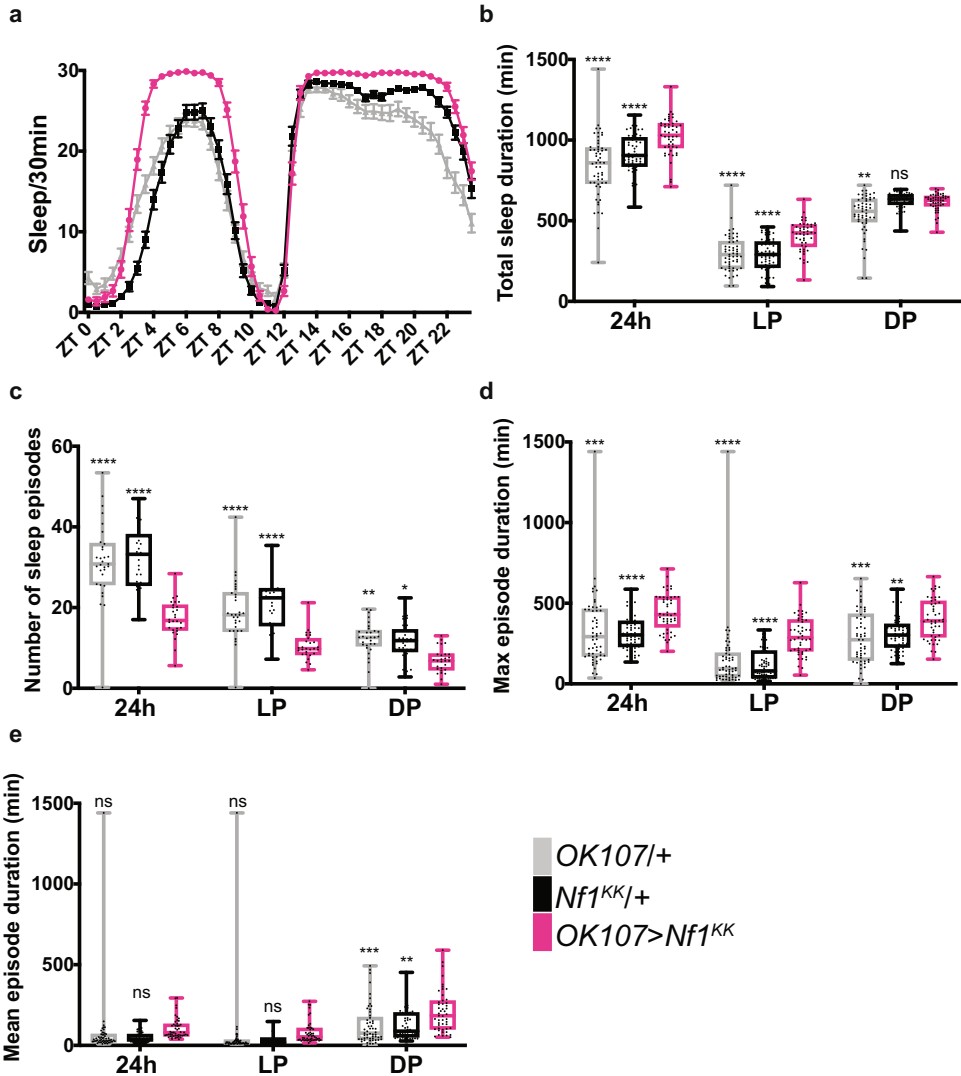

**Fig. 4 NF1 in the MB promotes daytime wakefulness. a** Sleep profiles of *OK107 > Nf1^KK^*, GAL4-only control (*OK107/+*) and UAS-only control (*Nf1^KK^/+*) flies per 30-min bin averaged over 5 days during LD (n = 49-59 per group). Values represent the mean ± SEM. **b–e** Different sleep variables were analyzed over a 24 h period, during the light phase (LP) and during the dark phase (DP). The center lines of the boxplots indicate the median, box boundaries are 25th and 75th percentiles, and the whiskers represent the minimum and maximum values. Values of *OK107 > Nf1^KK^* were compared with those of *OK107/ +* and *Nf1^KK^/+* using the two-way ANOVA with Dunnett's multiple comparisons test (\*p < 0.05, \*\*p < 0.01, \*\*\*p < 0.001, and \*\*\*\*p < 0.0001). **b** Total sleep duration. **c** Number of sleep episodes. **d** Maximum sleep episode duration. **e** Mean sleep episode duration. Source data are provided as a Source Data file.

and the reduction in the number of sleep episodes during the daytime (Fig. 6a–e). Furthermore, expression of a dominant-negative form of the PKA regulatory subunit PKA-R1(BDK)[62] using OK107 also increased daytime sleep (Fig. 6f–j). Neither *Pka-C1* RNAi nor PKA-R1(BDK) affected circadian locomotor activity when expressed in the MB (Supplementary Fig. 3a). Activity levels during the wake was also not significantly different between controls and *Pka-C1*-knockdown (Supplementary Fig. 3b). These results indicate that the activity of endogenous PKA-C1 in the MB promotes daytime wakefulness.

Our observation that *Nf1* and *Pka-C1* are both expressed in the MB and mediate daytime wakefulness are in accordance with the idea that PKA-C1 acts downstream of NF1. NF1 appears to control cAMP homeostasis through both Ras-dependent and -independent modes[56,59]. Thus, we sought to test whether Ras signaling mediates the effect of MB NF1 on sleep. Since expression of the constitutively active form of Ras, Ras^V12^, driven by OK107-GAL4 was lethal, we conditionally expressed it during adulthood using MB-GS (Supplementary Fig. 4a–e). This

manipulation resulted in an overall increase in the mean sleep duration and consolidation, although to a much lesser extent than NF1 or PKA-C1 knockdown and PKA-C1-dominant-negative expression, and not specific to daytime. Taken together, these results suggest that NF1 in the MB promotes wakefulness, at least partly, by a Ras-independent mechanism through the activation of cAMP–PKA signaling.

**cAMP and calcium rhythms in MB neurons mediate circadian wake drive.** How do NF1 and PKA-C1 in MB neurons promote daytime wakefulness? Previous studies have shown that PKA activity and sleep are inversely correlated[29,46] and cAMP–PKA activity increases intracellular calcium levels in MB neurons[63]. Therefore, we hypothesized that (i) cAMP and calcium levels in the MB exhibit circadian rhythms that peak during the daytime, which promote wakefulness during the day; and (ii) NF1 and PKA-C1 mediate these rhythms. To test these hypotheses, we first monitored cAMP levels in MB neurons using Epac1-camps, a

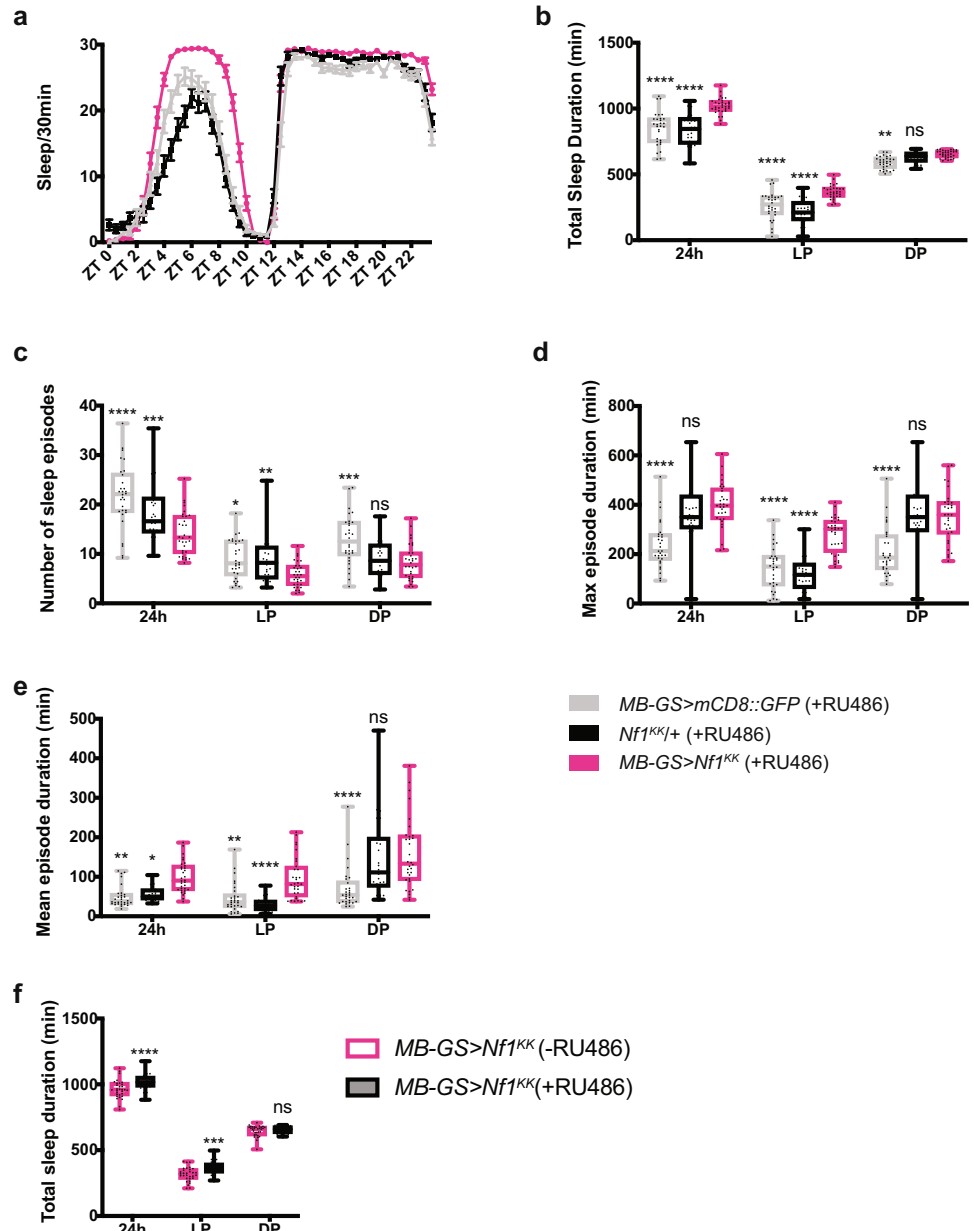

**Fig. 5 NF1 in adult MB promotes daytime wakefulness. a** Sleep profiles of *MB-GS > NF1^KK*, *MB-GS > mCD8::GFP*, and *Nf1^KK/+* fed with RU486 after eclosion to selectively induce MB-GS activity during adulthood. Sleep per 30 min bin averaged over 5 days of LD in the presence of RU486 is shown (*n* = 29–32 per group). Data are represented as mean values ± SEM. **b–e** Different sleep variables analyzed over 24 h, during the light phase (LP) and the dark phase (DP). The center lines of the boxplots indicate the median, box boundaries are 25th and 75th percentiles, and the whiskers represent the minimum and maximum values. Values of *MB-GS > NF1^KK* were compared with those of the two controls by two-way ANOVA with Dunnett's correction for multiple comparisons (*$p < 0.05$, **$p < 0.01$, ***$p < 0.001$, and ****$p < 0.0001$). **b** Total sleep duration. **c** Number of sleep episodes. **d** Maximum sleep episode duration. **e** Mean sleep episode duration. **f** Total sleep duration of the *MB-GS > NF1^KK* flies fed with RU486 (+RU486) or those without RU486 (−RU486). *n* = 32 for +RU486 and *n* = 32 for -RU486. Two-way ANOVA with Dunnett's correction for multiple comparisons found an increase in daytime sleep and sleep over a 24 h period following the induction of MB-GS with RU486 (***$p < 0.001$ and ****$p < 0.0001$). Source data are provided as a Source Data file.

genetically encoded cAMP sensor[64]. The rise in intracellular cAMP levels disrupts fluorescence resonance energy transfer (FRET) from CFP to YFP of Epac1-camps, which increases the ratio of CFP/YFP emission. We drove *Epac1-camps* with *30Y-GAL4*, which is expressed in all MB lobes[35], and imaged the brains dissected at 6 h intervals in LD and on DD2. In control (*w^1118*) background, the CFP/YFP ratio in the MB exhibited rhythms that peaked at ZT0 in LD and at CT18 in DD (Fig. 7a, b). The CFP/YFP rhythms were abolished in *per^0* flies in DD

(Fig. 7c). It is notable that the cAMP rhythms are phase shifted by several hours between LD and DD, as is the case for *Pka-C1* RNA and protein rhythms (Figs. 2b, 3a, and 3b). These results indicate that cAMP levels in the MB are controlled by light and circadian clocks, and peak during the daytime.

We next monitored calcium levels in MB neurons using the CaLexA system, in which calcium-dependent nuclear import of the LexA-VP16-NFAT fusion transcriptional activator initiates the expression of a LexAop reporter. The CaLexA system in

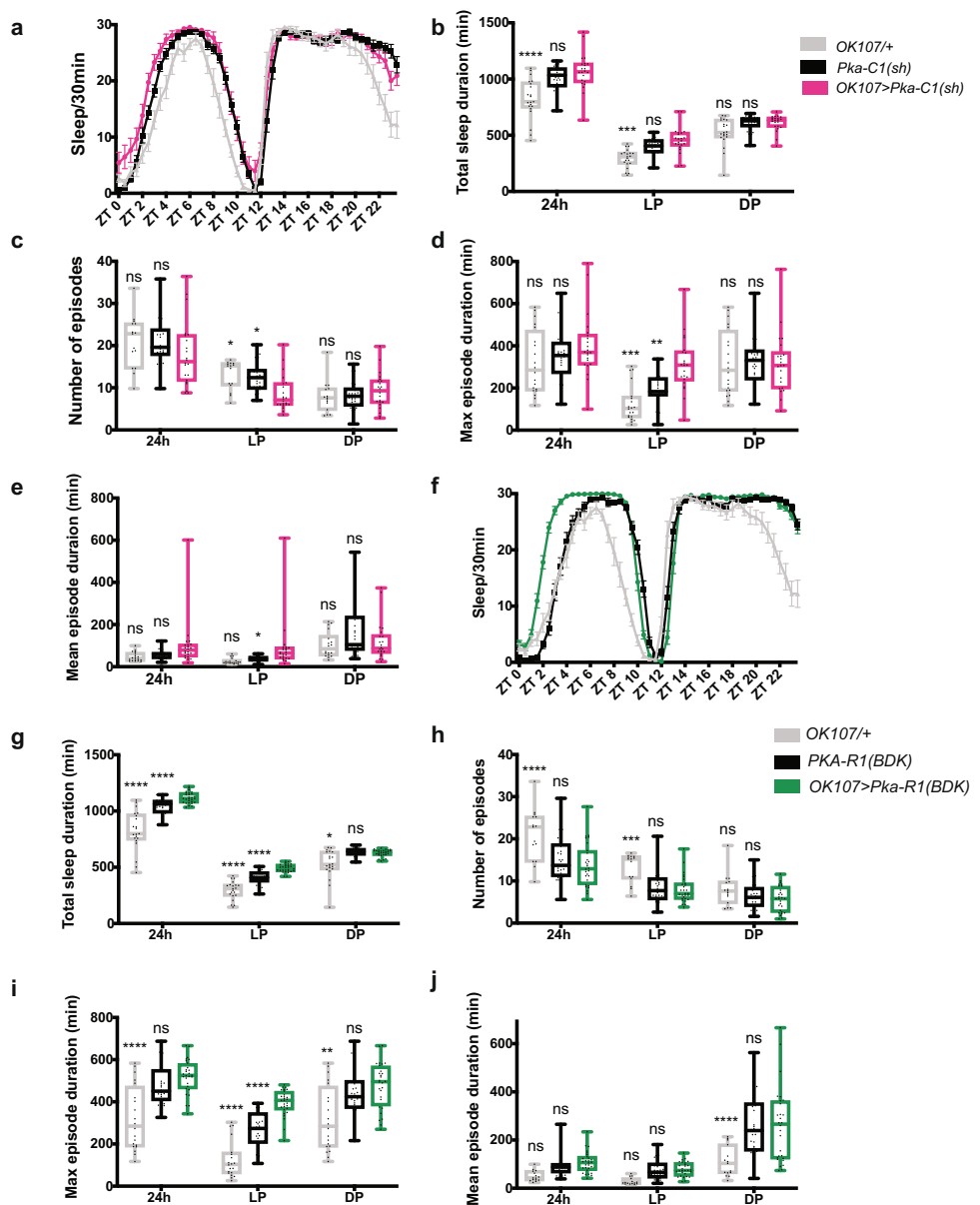

**Fig. 6 PKA-C1 activity in the MB promotes daytime wakefulness. a–e** Sleep profiles, amount, and structure of *OK107 > Pka-C1(sh)*, GAL4-only control (*OK107/+*), and UAS-only control (*Pka-C1(sh)/+*) flies (*n* = 21–32 per group). Values of *OK107 > Pka-C1(sh)* were compared with those of *OK107/ +* and *Pka-C1(sh)/+* using two-way ANOVA with Dunnett's multiple comparisons test (**p* < 0.05, ***p* < 0.01, ****p* < 0.001, and *****p* < 0.0001). The center lines of the boxplots indicate the median, box boundaries are 25th and 75th percentiles, and the whiskers represent the minimum and maximum values. **a** Sleep per 30 min bin averaged over 5 days during LD. Data are represented as mean values ± SEM. **b** Total sleep duration. **c** Number of sleep episodes. **d** Maximum sleep episode duration. **e** Mean sleep episode duration. **f–j** Sleep profiles (mean ± SEM) and analysis results of *OK107 > PKA-R1(BDK)*, *OK107/+*, and *PKA-R1(BDK)/+* flies (*n* = 14–30 per genotype). The boxplots are as in **b–e**. Values of *OK107 > PKA-R1(BDK)* were compared with those of both controls using two-way ANOVA with Dunnett's multiple comparison's test (***p* < 0.01, ****p* < 0.001, and *****p* < 0.0001). Source data are provided as a Source Data file.

combination with a select GAL4 driver and LexAop-GFP reports the relative calcium levels in cells targeted by GAL4 as GFP intensity[65]. We drove CaLexA using the 30Y driver with and without *UAS-Nf1KK* or *UAS-Pka-C1KK*. The brains were co-immunostained for GFP and Fas II at 6 h intervals in LD and DD. The CaLexA-GFP levels were rhythmic in both LD and DD, which peaked during the day and subjective day, respectively (Fig. 8a, d–g). In the presence of *Nf1* or *Pka-C1* RNAi, the CaLexA signal was significantly reduced across nearly all timepoints in both LD and DD, but the morphology of the MB was unaffected (Fig. 8b–g). Residual circadian rhythmicity in the CaLexA-GFP signal was present with *Nf1*

knockdown (Supplementary Fig. 5a, c), but *Pka-C1* RNAi disrupted the CaLexA rhythms into ~12 h oscillations (Supplementary Fig. 5b, d). Importantly, the CaLexA-GFP levels were reduced across all timepoints and rhythms were completely abolished in *per0* flies in DD (Fig. 8h). These results demonstrate that MB neurons exhibit circadian rhythms in calcium levels, which are controlled by circadian clocks and mediated by *Nf1* and *Pka-C1* expressed in the MB. The finding that *Pka-C1* knockdown perturbs calcium rhythms of the MB more strongly than *Nf1* knockdown is congruent with the notion that PKA-C1 acts downstream of NF1. Taken together with the results of cAMP imaging and sleep assays, these data

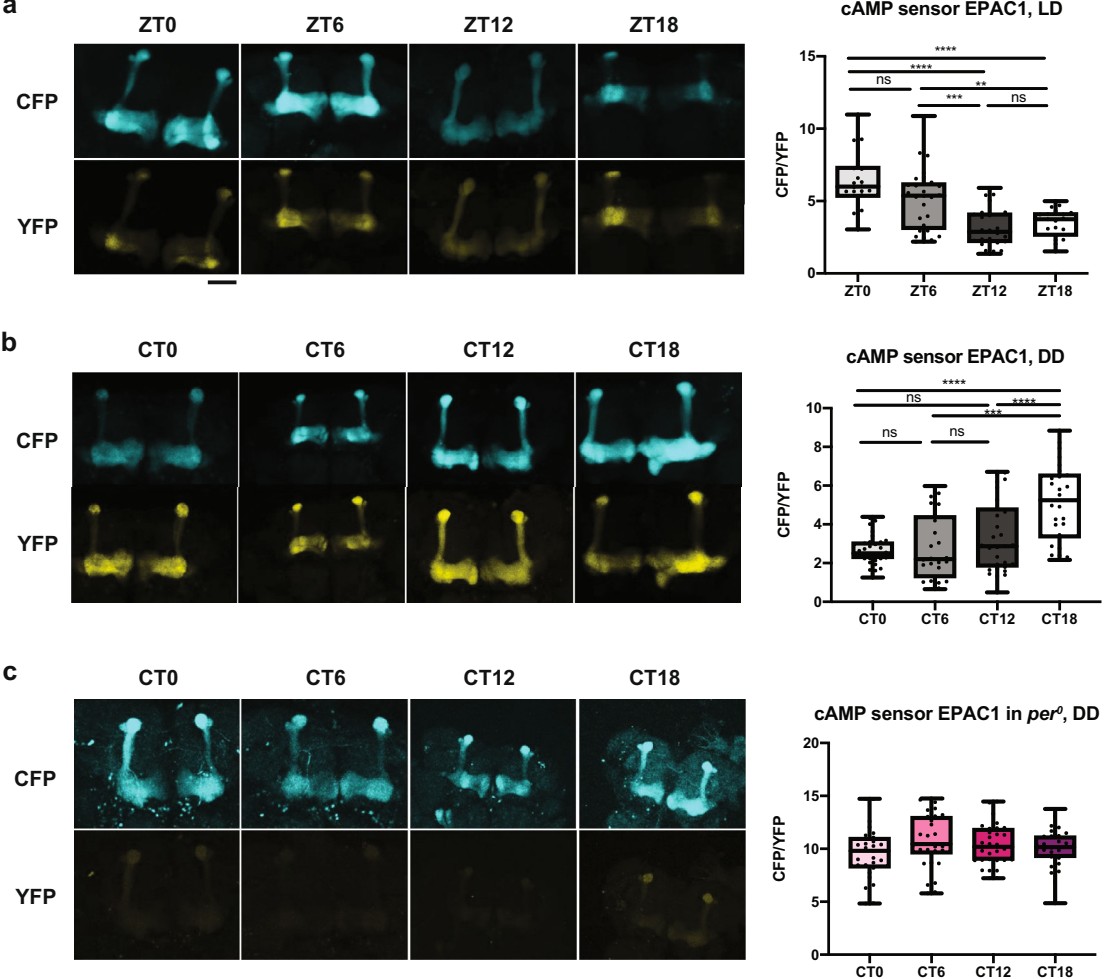

**Fig. 7 Circadian control of cAMP levels in the MB. a–c** Epac1-camp imaging of relative cAMP levels in MB neurons. Left, representative confocal images of the CFP and YFP emission. Scale bar, 25 μm. Right, quantification of the CFP/YFP ratio. The center lines of the boxplots indicate the median and the whiskers represent the minimum and maximum values. Values between timepoints were compared using one-way ANOVA with Tukey's multiple comparisons test (\*\**p* < 0.01, \*\*\**p* < 0.001, and \*\*\*\**p* < 0.0001). **a** Epac1-camp driven with *30Y-GAL4* in the *w*[1118] background. Images were acquired at four timepoints during LD. *n* = 20–30 brains per timepoint. **b** *w*[1118] in DD. *n* = 25–30 brains per timepoint. **c** *per*[0] in DD. *n* = 27–30 brains per timepoint. Source data are provided as a Source Data file.

indicate that circadian clocks increase MB excitability during the daytime via activating cAMP–PKA signaling, thereby promoting daytime wakefulness.

**Role of α/β and γ lobes in daytime wakefulness promotion by the NF1–cAMP/PKA pathway.** Distinct regions of the MB have been shown to be either sleep- or wake-promoting when stimulated individually[66], whereas the activation of the entire or subsets of KCs results in a compound effect on sleep that does not appear to be the sum of the individual effects[28–30]. Our findings that NF1 increases calcium levels across all MB lobes (Fig. 8b, d and f) and the propensity for wakefulness (Figs. 4 and 5) prompted us to examine whether the wake-promoting role of NF1 is localized to specific MB subregions. To this end, we drove *UAS-Nf1* RNAi with a set of GAL4 drivers expressed in different subsets of KCs[35] (Fig. 9a, c). Sleep was affected both positively or negatively by a variety of these drivers, consistent with the notion that the MB contains both sleep- and arousal-promoting neurons. Notably, however, the increase in daytime sleep, the most conspicuous phenotype induced by NF1 knockdown in the entire MB, was recapitulated with most of the drivers. These include c739, which is expressed mainly in

the α/β lobe. The increase in daytime sleep was most pronounced with 201Y, which is expressed in the α/β and γ lobes. The effect of NF1 knockdown only in the γ lobe was ambiguous, as the experiments with two γ lobe drivers gave inconsistent results (Fig. 9a, c).

To assess whether NF1 exhibits its wake-promoting role in these MB regions via activating the cAMP–PKA pathway, we drove *UAS-Pka-C1*[KK] with c309, 201Y, and c739 drivers. PKA-C1 knockdown restricted to the α/β lobes by using c739 did not consistently increase daytime sleep, whereas PKA-C1 RNAi driven with the c309 and 201Y drivers significantly increased daytime sleep (Fig. 9b, c). Taken together, these results suggest that the overt effect of the NF1–cAMP/PKA pathway on daytime wakefulness seems to emerge from simultaneous activation of α/β and γ KCs, without excluding the potential contribution of the α'/β' lobes.

## Discussion

Here we employed circadian RNA-seq analysis and identified a large number of genes expressed with diurnal and circadian rhythms in MB neurons, which are non-cell-autonomously driven by circadian clocks and regulated by light. Circadian clocks

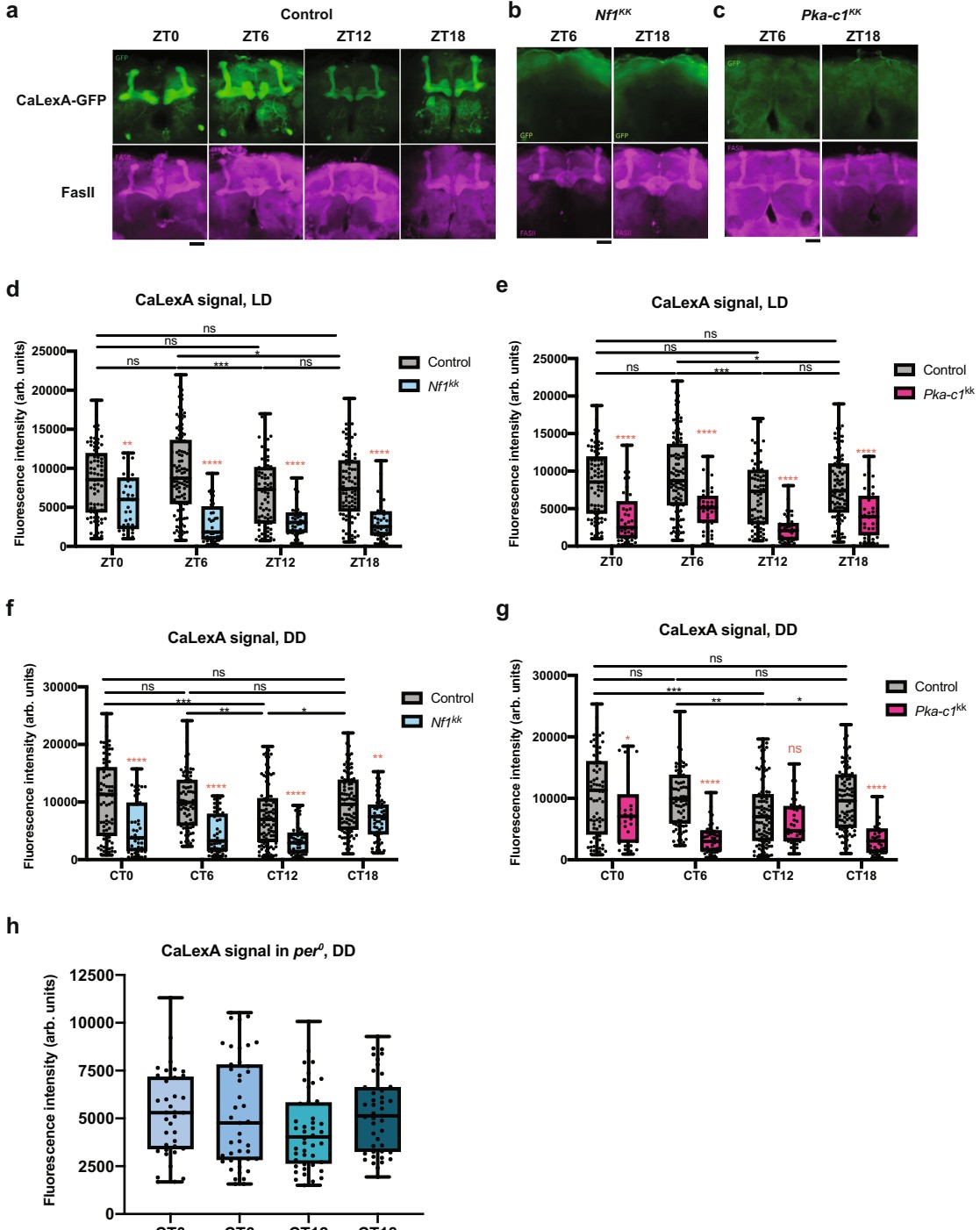

**Fig. 8 Circadian clocks control calcium rhythms in MB neurons via NF1 and PKA-C1.** Calcium-dependent expression of GFP in the MB lobes was monitored by driving CaLexA-GFP with *30Y-GAL4*, with or without *UAS-Nf1KK* or *UAS-Pka-C1KK* and co-staining with anti-GFP and anti-Fas II antibodies. **a** Representative confocal images of CaLexA-GFP expression monitored at four timepoints in LD in the control genotype. Scale bar, 25 μm. **b**, **c** Representative images of the CaLexA-GFP signal with the presence of *Nf1* RNAi (**b**) and *Pka-C1* RNAi (**c**). Scale bars, 25 μm. **d–g** Quantification of GFP intensity across all MB lobes in LD with or without *Nf1* RNAi (**d**), in LD with or without *Pka-C1* RNAi (**e**), in DD with or without *Nf1* RNAi (**f**), and in DD with or without *Pka-C1* RNAi (**g**). n = 45–60. arb. units arbitrary units. The center lines of the boxplots indicate the median and the whiskers represent the minimum and maximum values. Black stars indicate statistical significance levels comparing values within the control group between timepoints using the nonparametric Kruskal–Wallis one-way ANOVA with Dunn's multiple comparisons test (*p < 0.05, **p < 0.01, and ***p < 0.001). Red stars indicate the statistical significance levels comparing the control and knockdown groups at the same timepoint. (**p < 0.01, ***p < 0.001, and ****p < 0.0001, two-way ANOVA with Sidak's multiple comparisons test). Statistical test results comparing values between timepoints within the knockdown group are displayed in Fig. S5. **h** Quantification of GFP intensity across all MB lobes in DD in *per0*. The boxplots are as in **d–g**. n = 40–45. The Kruskal–Wallis one-way ANOVA with Dunn's multiple comparison's test found no significant difference between timepoints. GFP levels in *per0* were significantly lower than those in controls (displayed in **f** and **g**) in DD across all timepoints (****p < 0.0001 at CT0, 6, and 18, (**p < 0.01 at CT12, with the Kruskal–Wallis one-way ANOVA followed by Dunn's multiple comparisons test). Source data are provided as a Source Data file.

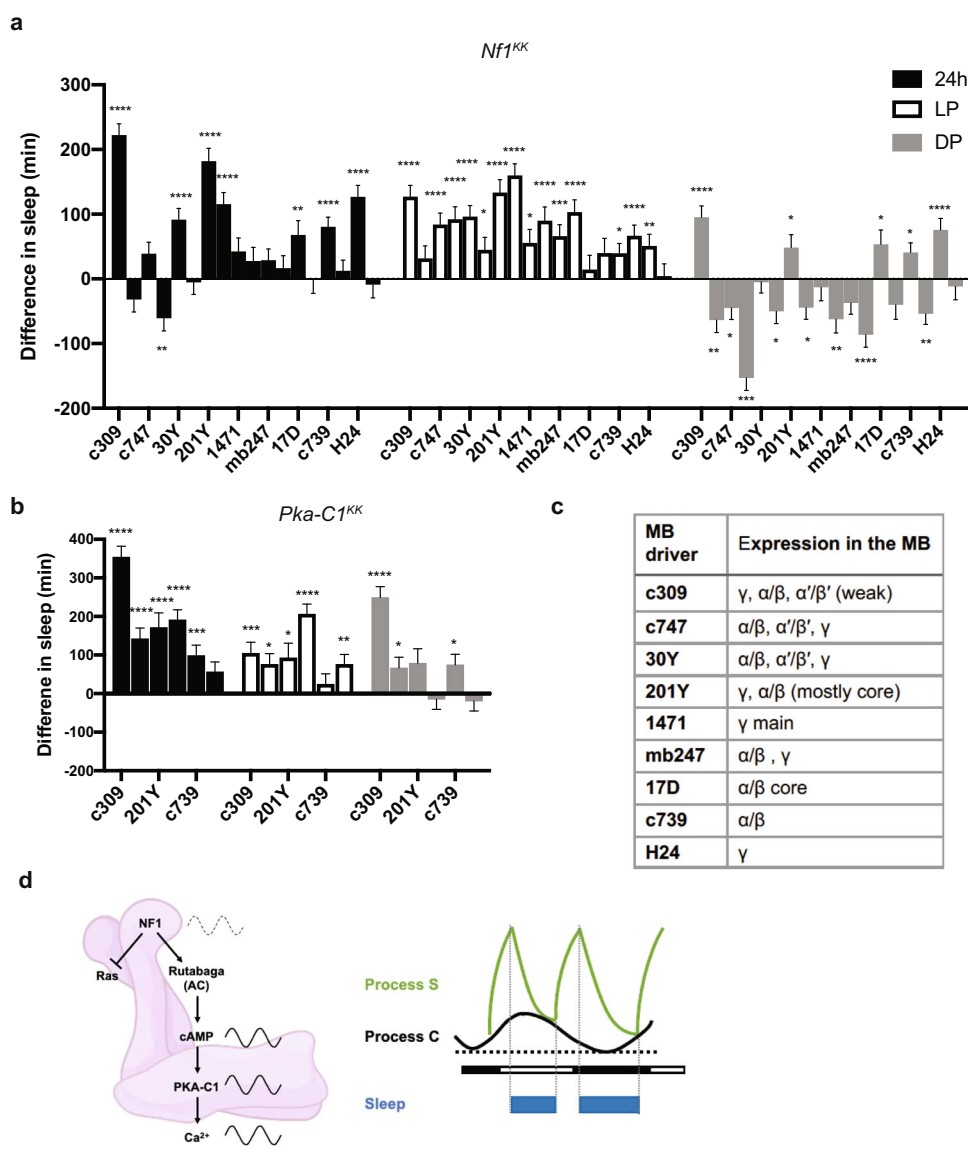

**Fig. 9 Role of the MB α/β and γ lobes in daytime wakefulness mediated by the NF1 and PKA-C1. a, b** Effects on sleep induced by *Nf1* RNAi (**a**) or *Pka-C1* RNAi (**b**) driven by different MB drivers. The plot represents sleep gain or loss in minutes of the experimental group (*GAL4 > UAS-RNAi*) as compared with the first (driver/+, left column) or second (*UAS-RNAi/+*, right column) control group. All values represent the mean ± SEM. Statistical analysis was performed on raw values using two-way ANOVA with Dunnett's multiple comparisons test (*$p < 0.05$, **$p < 0.01$, ***$p < 0.001$, and ****$p < 0.0001$; $n = 26-87$ flies per group). Source data are provided as a Source Data file. **c** List of MB drivers and their expression patterns. **d** A model of the role of NF1 in promoting daytime wakefulness. Left, NF1 activates PKA via Ras-dependent and -independent pathways. The Ras-independent pathway likely involves activation of adenylyl cyclase rutabaga. *Nf1* RNA is rhythmically expressed in MB neurons and thus NF1 protein levels might cycle as well (dotted wave indicate the hypothetical rhythms). cAMP levels in MB neurons are higher during the daytime than at night. PKA-C1 levels are higher during the daytime than at night, which gates the peak of calcium rhythms during the daytime. Thus, NF1 cascade activation increases neuronal excitability and calcium levels in MB neurons in a rhythmic fashion. Right, the activity of NF1 and cAMP−PKA signaling are higher during the daytime than at night. This conveys the circadian wake drive, known as Process C, in the two-process model of sleep regulation. Process C is controlled by the circadian oscillator and unaffected by the sleep. Homeostatic sleep drive (sleep debt), Process S, increases during wakefulness and declines during sleep. Since flies display bimodal activity patterns, fly Process S is assumed to be bimodal[101]. Process S triggers sleep near its peak and induces awakening at its trough. Circadian wake drive offsets the increase in sleep debt build-up. Knockdown of NF1 or PKA-C1 decreases circadian wake drive, thereby increasing the propensity of sleep during the daytime. Knockdown does little to nighttime sleep, since wake drive is already low (black dotted line).

also drive cAMP and calcium rhythms in MB neurons, tuning the MB activity higher during the day than at night. Some of the genes expressed with circadian rhythms in the MB, including *Nf1* and its downstream effector *Pka-C1*, promote wakefulness. *NF1* through activating cAMP–PKA signaling regulates calcium rhythms of MB neurons and gates wake drive during the daytime. These findings generally support the two-process model of sleep

regulation and suggest that Process C (circadian wake drive) is partly mediated in the MB by the NF1–cAMP/PKA pathway[34] (Fig. 9d).

**Circadian gene expression in non-clock neurons.** We show that 832 genes are expressed rhythmically in LD and 1144 genes cycle

on the second day during DD in MB neurons (Fig. 1b–d). The absence of rhythmic gene expression in MB neurons of *per⁰* flies demonstrate that these rhythms are driven by circadian clocks (Fig. 1f). These numbers, i.e., nearly 10% of the genes in the *Drosophila* genome cycle is surprisingly close to the percentage of cycling genes in a given clock-containing cell across metazoan[67–69]. These results demonstrate free-running gene expression rhythms in cells without canonical molecular clocks. Circadian RNA-seq analysis in non-clock-containing TH-GAL4-labeled dopaminergic neurons has been reported, but only LD timepoints were analyzed, showing ~100 cycling genes[70]. This unexpected difference in the number of cycling genes between MB neurons and dopaminergic neurons suggests that the physiology of the MB is tightly controlled by the clock neurons, highlighting its significance for the correct functioning of the MB. Our data also inform omics studies that time-of-day differences should be considered even when targeting non-clock cells.

Only ~15% of LD cyclers and ~10% of DD cyclers are rhythmically expressed under both LD and DD conditions. This relatively small overlap, in addition to the marked difference in phase distribution between LD and DD, indicates a strong control of gene expression in the MB by light (Fig. 1b−e). These findings are congruent with the notion that the MB is directly connected to multiple primary sensory processing centers including the optic lobe[71,72]. The MB integrates multimodal sensory inputs, including time of day, to coherently perceive the external environment. Our results suggest that gene expression changes within the MB driven by sensory information may be part of information processing.

Approximately 25% of the MB LD-DD cyclers tested in the mini-RNAi screen were found to be involved in sleep regulation (Fig. 2a and Supplementary Fig. 1a). The screen was by no means exhaustive, since only one RNAi line was used per gene, only a small fraction of the genes was tested, and some of the experiments were not replicated due to its exploratory nature. The effect of RNAi was not consistent in four genes between males and females, which may have been due to insufficient replicates. Nonetheless, the list of cycling genes established in the present study provides a valuable resource for the identification of genes involved in circadian or diurnal modulation of cognitive and behavioral states.

How can gene expression rhythms be driven in the cells that do not contain circadian clocks? Enrichment of binding sites for transcription factors and circadian-relevant miRNAs in cycling genes suggests that both transcriptional and miRNA-mediated mechanisms may be involved (Supplementary Table 2 and 3). The possibility of other post-transcriptional mechanisms, such as splicing, can be further tested gene-by-gene using qPCR or by sequencing total RNA in future studies. The relatively small number of enriched TFBSs could be due to the size of the input dataset (117 genes) as well as the heterogeneity of KCs. It should be noted that, several enriched TFBSs are the targets of TFs known to be expressed and to function within the MB. *Broad (br)* is expressed in the adult MB[73], specifically within the α/β lobes[74]. The expression of *forkhead (fkh)* and *ultraspiracle (usp)* in the adult MB have also been shown[75,76]. A previous study reported the enrichment of mRNA encoding *dorsal (dl)* in the α/βp MB neurons[74]. Overexpression of *dl* in the MB leads to fewer KCs and defasciculation of their axons[77]. Interestingly, *cactus* mRNA, which encodes a cofactor of Dorsal, is among our dataset of genes that cycle in the MB in both LD and DD (Supplementary Data 1). This suggests a possibility that a Cactus/Dorsal complex accumulates rhythmically in the MB, thereby controlling rhythmic expression of its target genes.

Some of the TFBSs identified in the cycling gene promoters did not reach the Z-score or Fisher-score cutoffs. This could be due to

the heterogeneity of KCs, and they may nevertheless regulate rhythmic gene expression in a subset of KCs. For instance, Chorion factor 2 (Cf2) has been shown to rhythmically drive the expression of the microRNA, mir-276a, which in turn binds to the 3'UTR of *timeless* and inhibits the translation of TIM at the proper time of day[78]. A similar mechanism may take place during the course of *Nf1* and *anne* expression, since they are targets of mir-276a (Supplementary Table 3).

Our results suggest that input from the circadian system provides temporal control of these transcriptional and/or post-transcriptional mechanisms. Which cells are afferent to the MB and how they transmit time-of-day information to modulate gene expression in KCs remain important open questions. A previous study used GRASP (GFP Reconstitution Across Synaptic Partner)[79] to demonstrate the physical contact between s-LNv pacemaker neurons and KCs[33]. Furthermore, activation of the PDF-positive s-LNvs was shown to induce calcium increase in the OK107-positive MB neurons[80]. These findings suggest a monosynaptic excitatory connection between the s-LNvs and KCs. The circadian system may also communicate with the MB via indirect synaptic inputs, volume transmission, and long-range signaling with neuropeptides. Whole-brain connectomics[81] may provide important information to address these possibilities.

**Role of NF1 and PKA-C1 in sleep.** We demonstrate that *Nf1* in the MB is wake-promoting. Previous studies have shown an opposite, sleep-promoting role of ubiquitously expressed *Nf1*[51,52]. Since we used the same *Nf1ᴷᴷ* RNAi strain as in their studies, the phenotypic differences are not experimental artifacts but indicate that NF1 can be both wake- and sleep-promoting depending on the cell type. The bidirectional regulation of sleep may be achieved by activating or inhibiting wake- or sleep-promoting neurons. More surprisingly, whereas we found that NF1 activates MB neurons (Fig. 8b, d and f), the Sehgal group showed that NF1 inhibits the activity of peptidergic neurons[49] using the same functional calcium imaging method. Therefore, NF1 downstream pathways also seem to diverge depending on the cell type.

In the MB, knockdown of NF1 or its downstream effector PKA-C1 reduces the calcium concentration and disrupts its rhythms (Fig. 8). Both circumstantial and experimental evidence from previous studies support these findings. *Nf1* deficiency reduces cAMP levels in *Drosophila* embryonic brains[58] and in murine astrocyte cultures[82]. Fly NF1 activates AC1 encoded by *rutabaga*[57,59], which is initiated by its interaction with the C-terminal domain of NF1[56]. Moreover, knockdown of *rutabaga* with *OK107-GAL4* reduces the nicotine-evoked $Ca^{2+}$-response in the MB, whereas *OK107 > Pka-C1 mc** has the opposite effect[63]. PKA likely phosphorylates $K^+$ channels[83,84], thereby rendering the cells more excitable. The fact that PKA activity is cAMP-dependent and cAMP levels are more elevated during the daytime than the nighttime (Fig. 7a) explains the daytime increase in calcium levels in the MB (Fig. 8a, d and e).

A previous study showed that the expression of *UAS-Pka-C1 mc** is sleep-promoting when driven by *201Y-GAL4* but wake-promoting when driven by *c309-GAL4*[29]. *201Y* is mostly expressed in the α/β core and γ neurons, whereas *c309* expression covers all MB lobes, similar to *OK107*. A functional calcium imaging study reported that calcium levels increase broadly in the MB when flies are awake and decrease with sleep[85]. In contrast, thermogenetic activation of individual subsets of KCs using split-GAL4 drivers has shown that α'/β' and γ main neurons are wake-promoting and γ dorsal neurons are sleep-promoting[66]. Therefore, although individual KC subtypes can be sleep- or wake-promoting when stimulated separately, simultaneous and constitutive activation of most KCs causes a net increase in

wakefulness in baseline sleep. Our data that NF1 or PKA-C1 RNAi with 201Y or c309 drivers increases daytime sleep (Fig. 9a, b) are in accordance with this interpretation. It should be noted that although α/β and γ neurons are most relevant for the action of NF1 and PKA-C1, our data do not exclude the contribution of α'/β' KCs.

*Nf1* mutant flies have learning disabilities[86,87], reminiscent of one of the manifestations of NF1[88]. *Nf1*-dependent olfactory learning in flies involves the cAMP pathway and requires the expression of *Nf1* in the α/β lobes of the adult MB[48]. Our study shows a role for *Nf1* expressed in the MB in sleep regulation. Considering that patients with NF1 have a higher prevalence of sleep disturbances, including daytime sleepiness[89,90], the fact that *Nf1* in the same subset of MB neurons is necessary for both memory formation and sleep regulation in flies is remarkable. These findings also suggest that the study of the role of *Nf1* expressed in the MB may shed light on some aspects of NF1 pathology.

## Methods

**Fly strains**. All *Drosophila melanogaster* strains were maintained on standard cornmeal-agar food at 18 °C or 25 °C in a 12 h light–dark cycle and controlled humidity. Detailed information on the strains used in the present study is provided in the Supplementary Table 6.

**Brain dissection and collection of MB neurons**. To collect cells during the LD, 2- to 4-day-old adult flies carrying the *OK107-GAL4, UAS-mCD8::GFP* transgenes were entrained to LD cycles for 3 days and dissected on the 4th day of LD. DD samples from wild-type and *per⁰* flies were collected on the second day of DD following 3 days of entrainment to LD. At each timepoint, the brains of 25 flies, including both males and females, were dissected and triturated following the protocol described in[36,91]. Optic lobes were removed and discarded during dissection. The cell suspension was kept on ice in a low-binding Eppendorf tube and immediately processed for fluorescent-activated cell sorting (FACS). Sorting of the GFP-positive cells was performed at the Flow Cytometry Core Facility of the University of Geneva using a Bio-Rad S3 Cell Sorter. All procedures were performed using low-binding plasticware. Before sorting, the cell suspension was filtered through a Filcon 70 μm strainer (BD biosciences, Cat. # 340606). Roughly 27,000 GFP-positive cells (equivalent to ~350 μL) were collected into 500 μL Buffer RLT Plus (QIAGEN RNeasy Plus Micro Kit, Cat. # 74034). Midway through the sorting process, the machine was halted in order to vortex the suspension containing the first ~13,500 collected cells. This ensures the proper lysis of the cells and the collection of cell-containing droplets that may be on the walls of the centrifuge tube. The collection tube was vortexed once more at the end of the sorting. Cells and cell lysates were kept on ice during all steps. Total RNA extraction was performed according to the manufacturer's protocol (QIAGEN RNeasy Plus Micro Kit, Cat. # 74034) and eluted in 14 μL supplied elution buffer. From the ~12 μL of total RNA collected, 1.5 μL was transferred into a separate tube to test RNA integrity and concentration. All samples were flash-frozen in liquid nitrogen and stored at −80 °C until further processing.

**RNA-seq of isolated MB neurons and data analysis**. All steps required for the RNA integrity check, library preparation and sequencing were performed by experts at the iGE3 Genomics Platform at the University of Geneva. Total RNA integrity and quantitation was performed using the Bioanalyzer High Sensitivity RNA 6000 Pico kit (Agilent, Cat. # 5067-1513). cDNA library preparations were conducted using the SMART-Seq v4 Ultra Low Input kit from TaKaRa (Cat. # 634894) according to the manufacturer's instructions. Libraries were sequenced on the HiSeq2500 Illumina machines using single-end 50-bp reads to generate more than 10 million reads per library. The iGE3 Genomics Platform performed QC, trimming of sequencing adaptors, and filtering of low-quality reads. Raw reads were aligned with the Dm3 *Drosophila* genome using the EPFL's HTSstation platform pipeline[92]. Because HTSstation platform has been discontinued, for *per⁰* RNA-seq data analysis, raw reads were aligned with the Dm6 Drosophila genome using STAR software (v2.7.9a). The raw counts of reads associated with each of the exons were determined using featureCounts function of the Rsubread package (v1.22.2). All libraries generated on average ~18 million single-end 50 bp reads, from which approximately 80% (~14 million reads) were mapped to the genome. The normalized expression levels (Reads Per Kilobase of Million mapped reads; RPKM) were used for downstream analysis.

**Analysis of circadian gene expression**. Genes with an expression <1 RPKM at any timepoint were excluded from the circadian analysis. Expression profiles were analyzed for rhythmicity in R (r-project.org) using JTK_CYCLE (v3.1)[37]. JTK_CYCLE algorithm uses a nonparametric Jonckheere-Terpstra test with the

Bonferroni correction. RNA expression profiles and heatmaps were produced using ggplot2 (v3.1.0) in R. The significance threshold was set to an adjusted *p*-value (*p.adj*) < 0.05.

Gene Ontology term enrichment analysis of cycling genes was performed using Gorilla (http://cbl-gorilla.cs.technion.ac.il/)[93]. Transcription factor binding sites (TFBS) within the ±2 kb of the transcription start site were identified using the oPOSSUM3 tool (http://opossum.cisreg.ca/oPOSSUM3/)[41]. miRNA target site analysis was performed using the microRNA.org database (http://www.microrna.org/)[44,45].

**Sleep assays and analysis**. The fly sleep assay was performed using the Drosophila Activity Monitoring (DAM) system (Trikinetics) as described previously[94]. Briefly, 2-to 5-day-old male or virgin female flies were placed individually in 65 × 5 mm glass tubes containing 5% agarose with 2% sucrose. To induce MB-GeneSwitch expression, 200 μg/mL RU486 (Mifepristone; Sigma, Cat. # M8046) was added. Flies were entrained for 3 days in LD (12 h light–dark), and data for sleep analysis were collected during the following 5 days of LD. Locomotor activity was collected in 1 min bins, from which sleep quantity, defined as bouts of inactivity lasting for 5 min or longer[95,96], was extracted. All sleep behavior assays were performed at 25 °C.

Sleep analysis was performed in MATLAB (MathWorks, version R2017a, build 9.2.0.538062) using SCAMP (v2) (Vecsey laboratory)[97] according to the instruction manual. The raw data obtained with SCAMP were compared using ANOVA followed by Dunnett's post-hoc tests for multiple comparisons in GraphPad Prism (v.8.1). Unless otherwise indicated, all sleep data were analyzed from experiments using virgin female flies and plots were generated in GraphPad Prism (v.8.1).

**Immunohistochemistry and microscopy**. All the imaging experiments were conducted at least in duplicate. GFP expression in the MB driven by the CalexA system was visualized using whole-mount brain immunofluorescence with anti-GFP and anti-Fasciclin II (Fas 2) antibodies following the previously described protocol[98] with minor modifications. Briefly, 5- to 6-day-old flies were decapitated at four different timepoints (ZT/CT0, ZT/CT6, ZT/CT12, and ZT/CT18) following 3 days of LD-entrainment for LD condition. For experiments in DD, the brains were collected in DD2 following the 3-day LD-entrainment. Then, the heads were fixed in 4% paraformaldehyde + 0.3% Triton X-100 for 1 h on ice and washed quickly twice and then twice for 20 min in PBST-0.3 (PBS, 0.3% Triton X-100). Subsequently, the head cuticle was partly removed, and the heads were washed twice more, blocked in blocking solution (55% normal goat serum, PBST-0.3) for 1 h at room temperature, and incubated with the primary antibodies for 48 h at 4 °C. Rabbit affinity purified anti-GFP (1:500, Invitrogen G10362) and 1D4 mouse anti-Fasciclin II (1:5; Developmental Studies Hybridoma Bank (DSHB)) (were used as primary antibodies, and Alexa 488 goat anti-rabbit IgG (1:250, Thermo Fisher A11008) and Alexa 633 goat anti-mouse IgG (1:250, Thermo Fisher A21052) were used as secondary antibodies. The remaining cuticle was removed from the head and the brains were mounted on slides with Vectashield (Vector Laboratories). Images were acquired using a Leica TCS SP5 confocal microscope.

PKA-C1 protein levels in the MB were measured using whole-mount brain immunofluorescence with anti-PKA-C1 and anti-Fas 2 antibodies as described above. 5- to 6-day-old flies were decapitated at 6 different timepoints (ZT/CT0, ZT/CT4, ZT/CT8, ZT/CT12, ZT/CT16, and ZT/CT20) as described above. The primary antibodies used were guinea pig anti-PKA-C1[99] (1:500, a gift from Dr. Pascal Thérond) and 1D4 mouse anti-Fasciclin II (1:5; Developmental Studies Hybridoma Bank (DSHB)).

**Epac1-camps imaging**. One- to Three-day-old flies were decapitated at four different timepoints (ZT/CT0, ZT/CT6, ZT/CT12, and ZT/CT18) following 3 days of LD-entrainment for LD experiments. For DD experiments, flies were kept one day in DD after entrainment before dissection. Imaging was performed as previously described in Shafer et al., (2008)[64]. The brains were dissected in Ringer's solution and mounted on slides with hemolymph like saline HL3 solution. Images were acquired using a Leica TCS SP5 confocal microscope. This experiment was not extended beyond an hour following dissection and mounting to avoid possible photobleaching and the disruption of the brains.

**Image analysis**. Confocal Z-stacks were analyzed using the Image J/Fiji software (v2.1.0/1.53c)[100]. GFP and PKA-C1 fluorescence intensities in the MB were measured from the sum slice Z-projections within the area defined by the Fas II signal. For Epac1-camps image analysis, CFP (480 nm) and YFP (530 nm) emission intensities in the MB were measured from the sum slice Z-projections.

**Statistics and reproducibility**. GraphPad Prism (v.8.1) was used for statistical analysis and data plotting. Normally distributed data were compared using parametric tests and non-normally distributed data were analyzed using nonparametric tests. Sleep data were analyzed using one- or two-way ANOVA with Dunnett's post-hoc multiple comparisons test. Statistical comparison of fluorescence intensities was performed using the ordinary one-way ANOVA with Turkey's multiple comparison's test when data were normally distributed, and the Kruskal–Wallis one-way ANOVA with Dunn's multiple comparison's test was used to compare non-normally distributed data. The two-way ANOVA with Sidak's multiple

comparisons test was used to compare GFP intensities between control and knockdown groups at the same timepoint in the CaLexA experiments. Significant values in all figures are: *$p < 0.05$, **$p < 0.01$. ***$p < 0.001$, and ****$p < 0.0001$. ns indicates not significant. All the experiment except for the fly sleep experiments presented in Fig. 2a and Supplementary Fig. 1a, b, were repeated at least twice. The genetic screens shown in the Fig. 2a and Supplementary Fig. 1a and b were not repeated because of the exploratory nature of these experiments. Micrographs are representatives of two or more independent experiments.

**Reporting summary**. Further information on research design is available in the Nature Research Reporting Summary linked to this article.

## Data availability

RNA-seq data are deposited in the Sequence Read Archive (SRA) (https://www.ncbi.nlm.nih.gov/sra) as a Bioproject PRJNA719003, with the accession numbers SRR14127352- SRR14127363, SRR14127675- SRR14127686, and SRR14213543-SRR14213554. Detailed information of the RNA-seq samples and their corresponding accession numbers are included in the Source Data. The databases used in this study are microRNA.org (http://www.microrna.org/), oPOSSUM (v3.0) (http://opossum.cisreg.ca/oPOSSUM3/), and Gorilla (http://cbl-gorilla.cs.technion.ac.il/). Source data are provided with this paper.

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

## Acknowledgements
We thank Simon Sprecher for *MB-GS* line, and Bloomington Drosophila Stock Center and VDRC for other fly strains. We thank Dr. Pascal Thérond for anti-PKA-C1 antibodies. We also thank Dr. Giorgio Gilestro and Dr. Ueli Schibler for their helpful comments on this study. This research was funded by grants from the Swiss National Science Foundation (31003A_169548 and 310030_189169), the JST PRESTO program (10101675), and the Georges and Antoine Claraz Foundation. We are grateful for the support of the EuroScholars program to A.F. and the Plan Strategique Sciences Vie (PSVIE) of the University of Geneva to B.L. Some figures were created using BioRender.com.

## Author contributions
E.N. and P.M. and planned the experiments. P.M., B.L., L.S., and A.F. conducted the experiments. P.M., B.L., A.F., L.S., and E.N. analyzed the data. B.L., L.S., and E.N. wrote the manuscript.

## Competing interests
The authors declare no competing interests.
