## [Peer Review File · Nature Communications]

Reviewers' Comments:

Reviewer #1:

Remarks to the Author:

The manuscript from Almeida et al examines how circadian rhythms of behavior (sleep) emerge from the activity of a small set of clock neurons that communicate with the rest of the brain to control the behavior of the whole animal – in this case, in the fruitfly.

Almeida et al focus on mushroom body neurons, which have already been implicated by several other groups in control of sleep and activity. Their approach is innovative in that they examine RNA levels in the mushroom bodies and find a set of genes that are rhythmically expressed in light:dark cycles and in constant darkness. This is interesting because the mushroom bodies do not express clock genes, and the authors propose that these rhythms must be being driven by the circadian clock, especially since there are rhythms in constant dark conditions. Then they screen some of these mushroom body genes for altered sleep phenotypes and find many phenotypes. They focus on NF1 and PKA since these proteins fit into a single pathway.

However, I have a number of major problems with the study.

1. The OK107-Gal4 driver line is expressed in the pars intercerebralis in addition to the mushroom body. Images available online seem to also show expression in other regions of the central brain. If this is true, then the conclusions of the study are quite flawed. The authors can address this by showing rhythms of the key genes (Nf1 and Pka-C1) in mushroom body neurons independently of OK107-Gal4, using antibodies or ideally in situ hybridization. This is particularly important since some pars intercerebralis are directly downstream of clock neurons.

2. The authors conclude that the rhythms in RNA levels are controlled by the circadian clock. I agree that this seems likely since some genes show rhythms in DD, which is very interesting. However, the authors do not test if the rhythms depend on clock genes and/or clock neurons, both of which are relatively easily tested. It is quite conceivable that the rhythmic RNA levels are driven by the activity of the fly. Of course, this is controlled by the circadian clock, but this would be a very different conclusion than the one described in the manuscript.

3. The CaLexA experiment in Figure 6 is very interesting, but the FasII staining looks very different with Nf1 RNAi. How did the authors conclude that mushroom body morphology was “unaffected”? Do the authors have the spatial precision to conclude this? Also why change to 30Y as a driver for this CaLexA experiment?

4. Based on the slightly confusing set of drivers used in Figure 6C, where do the authors think that Nf1 is expressed?

Minor comments:

1. For the transcription factor motif study, are these factors expressed in the mushroom body? Are any of them rhythmically expressed? I am not sure if these data are useful in the manuscript.

2. As a non-sleep expert, are 1000 minutes of sleep (16.5hr) in control flies higher than normal?

3. The authors should cite Pirez et al (2019), which shows that s-LNvs signal to mushroom body neurons and this addresses their doubt about LNV-MB connectivity being functional (page 17 in their manuscript).

Reviewer #2:

Remarks to the Author:

The premise of this paper is interesting and the approach of RNA-seq validates the initial premise to a large dataset. Essentially, cells with circadian clock mechanisms drive cyclical gene expression of downstream genes, resulting in cell/tissue-specific sets of rhythmic RNA levels that are thought to make a significant contribution to specific rhythmic phenotypes in some aspects of cellular physiology, behavior etc. However, only a small subset of brain neurons have a clock mechanism

capable of directly driving endogenous rhythmic gene expression. If cycling mRNA levels is a major aspect of how cells/tissues/organs are regulated in a daily manner, is it possible that clock cells somehow drive cyclical gene expression in non-clock neurons to generate rhythmic behavior? The authors apply this strategy to the mushroom bodies (MBs), a "clockless" brain region critical to sleep, memory, etc, behaviors that are known to exhibit a circadian component. Indeed, they identify 100's of cycling mRNAs when probing isolated MBs, despite the absence of clock gene expression. Although prior studies have shown this possibility in other tissues, this study generates a very large dataset that can be mined for many uses. In this study the authors focus on a smaller subset of mRNAs that cycle in both LD and DD (117), leading them to test 21 by RNAi in the MBs. 5 of these led to increased sleep in males and females, and for the rest of the paper they focus on Nf1 (Neurofibromin 1) and Pka-C1. Prior work has already shown roles for Nf1 and Pka-C1 in circadian rhythms and sleep in *Drosophila*. The authors provide additional data that Nf1 effects on sleep in the MBs is at least partly due to controlling Pka-C1 activity. Where this manuscript runs into major trouble is with the over-interpretation of their findings. Strip that away and this paper provides; (1) a nice dataset of cyclical gene expression in non-clock containing neurons, and (2) insight into the previously identified MB-wake/sleep regulators Nf1 and Pka-C1, by showing that Nf1 partly functions via Pka-C1 activation.

Below are a few examples of the distance between the data and the claims?:

1. They never show that cyclical RNA expression in the MBs is dependent on functional clocks. While this might seem obvious, especially that some continue in DD, this was not formally shown.
2. More of a problem is the major claim (see title) that it is the "circadian" cycling of Nf1 and Pka-C1 that is critical to the rhythmic MB-dependent or -driven behaviors previously shown. There are numerous sub-issues within the veracity of such a claim; for example;
 - a. The authors never show that eliminating the cycling of endogenous Nf1 and/or Pka-C1 mRNA levels in the MBs eliminates a daily behavior. Sure this is very difficult to do but then the claim cannot be made. Overexpressing constitutively activated PKA does not do the trick.
 - b. Indeed, related to the above, they do not show that Pka-C1 protein levels (and especially) activity undergoes daily rhythms. The same is true for Nf1. This is further compounded by the admitted heterogeneity of the MBs and different functional outcomes depending on spatial concerns. For example, perhaps in the key MB cells driving rhythmic wake-sleep, Pka-C1 is not cycling or shows different phases compared to other regions. Not all cycling mRNAs lead to cycling protein levels. Clearly, the overall cycling of Pka-C1 is consistent with decreased sleep during the day as the authors noted, however....
 - c. More of a stretch has to be assumed by the authors in building their model in the case of Nf1, as it peaks in the end of the night...e.g., "Since Nf1 mRNA rhythms peak at the end of the night in LD (Figure 2B), NF1 protein levels likely peak several hours later during the daytime, driving higher wake propensity during the day than at night" (page 11). This begs for measuring Nf1 levels.
 - d. Wake-sleep cycles and the timing of behavior are relatively similar in LD and DD, especially after only 1 day of DD as in the RNA-seq protocol used herein. Yet, the mRNA cycling profiles of Pka-C1 and Nf1 are very different (fig. 2B). It is hard to understand how this reasonably explains circadian modulation of sleep drive (process C). This is even more confused by the heterogeneity in different parts of the MBs as either activating wake or promoting sleep. Moreover, especially in LD cycles, activity is bimodal. Yes, flies are more active during the day compared to night, but they have a morning and evening peak of activity. Wherein does a unimodal daily cycle in gene expression (e.g., Nf1, Pka-C1) drive a bimodal C-process in daily activity?
 - e. The authors did not eliminate a possible contribution to sleep homeostasis from Nf1 and Pka-C1? Just because its mRNA levels cycle does not mean it can only function in a circadian aspect. Alterations in S-process could appear as alterations in C and vice-versa.

Essentially, the authors did a very nice study, however they seriously over-interpreted their data to make quite fundamental claims. I recognize this is difficult terrain but if such claims are to be made they should be justified by the data presented. The evidence that cycling of Nf1 and Pka-C1, at the level of cycling mRNA abundance in the MBs, drives the circadian-dependent bimodal distribution of activity in *Drosophila* is not compelling. More fundamental insights would require, a) showing how clocks drive mRNA cycling in non-clock cells (speculation was put forth on transcription factors and miRNAs, but no evidence was presented), and/or b) show that the actual cycling on Nf1 and Pka-C1 in the MBs is required for the normally observed timing of daily activity in *Drosophila*; i.e., fundamentally tie the rhythmic gene expression to the rhythmic behavior.

Reviewer #3:

Remarks to the Author:

The manuscript by Nagoshi and colleagues identifies a significant problem defined by the identification of circadian pacemaker versus non-pacemaker regions of the brain: how is circadian information transmitted from the first to the second? How are specific brain regions circadian-rhythmic if they do not express robust clock proteins in a cell-autonomous fashion? The authors use molecular profiling of purified, genetically-identified neurons to demonstrate daily circadian cycling of ~100 transcripts in neurons that lack expression of canonical clock proteins (PER, TIM etc). These findings make clear that the PER-dependent clock is not required cell-autonomously to drive daily rhythmic gene expression.

The role of NF1 in regulating sleep is well-established so the significance lies centrally on whether there is increased clarification of the role of NF1 in KC (non-clock) cells.

A very recent Science paper by the Reddy lab demonstrates strong circadian control of gene expression in fibroblasts that lack the canonical molecular oscillator (Bmal k/o background). That suggests an ability of eukaryotic cells to generate daily rhythmic patterns without reference to the PER-dependent oscillator. At the least, the authors should acknowledge the possibility that KC cells of the fly brain may possess such abilities, and may therefore not require extrinsic timing information. To address this issue in greater experimental detail, they could knock down Clock or Per in the canonical pacemaker network and confirm continued rhythmic KC gene expression.

I have several questions about the behavioral studies in this study:

- (i) Behavior was tested in conventional fashion (locomotor assays) with 12-32 flies, but seemingly with an N=1; the authors should be clear on whether they tested genotypes once or more than once. From the description offered, it appears to have been once, which is not sufficient – especially with discrepant results between males and females.
- (ii) The authors created a list of 21 candidates to test based on each of several arbitrary criteria. This is a small list. More remarkable still, each candidate was tested only once (see point above) and apparently only with a single RNAi stock for each. In sum, my criticism is that this ‘testing’ represents a very narrow technical approach to the experimental question.
- (iii) When tested in males and females separately, some genes registered consistently different, but others not so – this was presented without comment or explanation. Perhaps some of these differences represent lack of sufficient replication.
- (iv) The results from similar experiments were presented in different formats (violin plots versus histograms): there was no explanation for doing so or what the different formats reveal.
- (v) The authors make the point that RNAi stocks tested for 5 of 20 LD-DD candidates affected sleep, whereas, 0 of 12 RNAi stocks for “LD-only” candidates affected sleep. They considered this a significant observation. My question is – on what basis can this be considered significant? But is there any basis to argue? These are extremely small numbers; Was this repeated? Is there a statistic that can be applied to make an argument for significance?
- (vi) I could not follow Figure S1 in which the same set of results were said to reveal significant differences (panel A) and no significant differences (panel B). More explanation would be appreciated to understand the nuances presented.

The CaLexa results are impressive. However, they are taken at a single arbitrary time point and show a small sub-region of the brain. I recommend displaying a few different time points and showing the whole brain, in addition to the magnified region that describes the KCs.

Argument that only a subset of KCs are affected appears not comprehensively-tested

There is no line specific for alpha prime/ beta prime.

In the absence of that, there is no logical basis to exclude a role for alpha prime/ beta prime KCs.

The previous studies of KC regulation of sleep revealed that there is both positive and negative regulation by different subsets. I was not completely persuaded the authors are adequately

representing their data to propose a model that is consistent with the previous studies. More clarity on this point is needed.

Response to Reviewers

Reviewer #1 (Remarks to the Author):

The manuscript from Almeida et al examines how circadian rhythms of behavior (sleep) emerge from the activity of a small set of clock neurons that communicate with the rest of the brain to control the behavior of the whole animal – in this case, in the fruitfly.

Almeida et al focus on mushroom body neurons, which have already been implicated by several other groups in control of sleep and activity. Their approach is innovative in that they examine RNA levels in the mushroom bodies and find a set of genes that are rhythmically expressed in light:dark cycles and in constant darkness. This is interesting because the mushroom bodies do not express clock genes, and the authors propose that these rhythms must be being driven by the circadian clock, especially since there are rhythms in constant dark conditions. Then they screen some of these mushroom body genes for altered sleep phenotypes and find many phenotypes. They focus on NF1 and PKA since these proteins fit into a single pathway.

Thank you for your constructive comments and suggestions. Below we summarize planned experiments to address them.

However, I have a number of major problems with the study.

1. The OK107-Gal4 driver line is expressed in the pars intercerebralis in addition to the mushroom body. Images available online seem to also show expression in other regions of the central brain. If this is true, then the conclusions of the study are quite flawed. The authors can address this by showing rhythms of the key genes (Nf1 and Pka-C1) in mushroom body neurons independently of OK107-Gal4, using antibodies or ideally in situ hybridization. This is particularly important since some pars intercerebralis are directly downstream of clock neurons.

Cells in the pars intercerebralis (PI) and central brain make up a very small proportion of the cells labeled by OK107-GAL4, compared to the MB intrinsic cells (Kenyon cells), which count approximately 2,000 cells per hemisphere. Therefore, it is not very likely that RNA-seq counts of cycling genes are entirely from non-MB cells. However, we agree that it is important to verify the rhythmic expression of Nf1 and Pka-C1 within the MB. We plan to address this issue first by

Experiment 1. Immunostaining of brains using Pka-C1 and NF1 antibodies at 6 timepoints in LD, DD and DD in clock-less *per0* background. Localization to the MB will be evaluated by co-immunostaining with anti-FasII antibody.

We have already obtained anti-PKA C1 antibodies, whereas it has been difficult to find antibodies that works for immunostaining of fly NF1. One antibody has been reported and used to detect expression of NF1 in the MB (Gouzi JY, et al. PLOS Genetics, 2011). However, this antibody was originally made in André Bernards' lab and published in 1993 (The et al, Science, 1997) and no longer available. To circumvent this problem, because fly NF1 protein is overall 60% identical to human's, we will test multiple antibodies against mammalian NF1 to find one that cross-reacts to fly NF1. We are cautiously optimistic that this approach will work out.

Experiment 2: In parallel, we will also examine temporal and spatial expression patterns of the protein encoded by several other MB-cycling genes by immunohistochemistry. Since it is widely known that not all cycling mRNAs produce cycling protein, it wouldn't be surprising that some of them are not expressed rhythmically in the MB. Whereas, a positive data that any of the tested cycling genes is rhythmically expressed in the MB would strengthen our findings.

2. The authors conclude that the rhythms in RNA levels are controlled by the circadian clock. I agree that this seems likely since some genes show rhythms in DD, which is very interesting. However, the authors do not test if the rhythms depend on clock genes and/or clock neurons, both of which are relatively easily tested. It is quite conceivable that the rhythmic RNA levels are driven by the activity of

the fly. Of course, this is controlled by the circadian clock, but this would be a very different conclusion than the one described in the manuscript.

Thank you for pointing out this important issue. We are setting up RNA-seq experiments to address this issue directly, as described below.

Experiment 3: We will isolate MB neurons of *per0* mutants at 6 timepoints in constant darkness and analyze their transcriptome using RNA-seq. This is a straightforward experiment in theory, but is a labor-intensive, time-consuming, technically non-trivial and costly experiment. However, since we have successfully completed circadian RNA-seq in the MB in LD and DD, we are confident that we could make it work.

The outcome of this new set of experiments will either demonstrate that transcriptome rhythms in MB neurons are non-cell-autonomously regulated by circadian clocks, or that MB transcriptome rhythms are generated by a clock-independent, as yet unknown mechanism. Either outcome will strengthen the significance of our paper.

3. The CaLexA experiment in Figure 6 is very interesting, but the FasII staining looks very different with Nf1 RNAi. How did the authors conclude that mushroom body morphology was “unaffected”? Do the authors have the spatial precision to conclude this? Also why change to 30Y as a driver for this CaLexA experiment?

We did many replicate experiments and confirmed that morphology of the MB is unaffected by NF1 knockdown. By mistake, we picked a picture that looks a bit deformed, perhaps because the brain was slightly twisted when mounted on the slide glass. We will replace it to a more representative micrograph in the revised manuscript.

We used 30Y-GAL4 because it is expressed in all MB lobes and is more practical for crosses. (OK107 is on the 4th chromosome. CaLexA-GFP consists of 3 transgenes. With GAL4 and UAS-RNAi, the lines required for this experiment have 4 or 5 transgenes.)

Additionally, as shown in the Figure A (displayed at the end of this letter), we have already performed CaLexA-GFP experiments at 4 timepoints in LD with and without NF1 knockdown. We found that calcium levels show daily rhythms with a peak during the daytime. NF1 knockdown significantly reduces the calcium levels at all timepoint, except at ZT12 when calcium levels in the wild-type is at its minimum. These results confirm that excitability of MB neurons is rhythmic and regulated by the NF1 within the MB.

4. Based on the slightly confusing set of drivers used in Figure 6C, where do the authors think that Nf1 is expressed?

The planned Experiment 1 will address answer this question.

Minor comments:

1. For the transcription factor motif study, are these factors expressed in the mushroom body? Are any of them rhythmically expressed? I am not sure if these data are useful in the manuscript.

As stated in Discussion (line 377-388), several transcription factors that are predicted to bind MB cycling gene promoters are known to be expressed and function in the MB. We also mention that *cactus*, which encodes the co-factor of one of the enriched transcription factors (Dorsal), is rhythmically expressed in the MB.

2. As a non-sleep expert, are 1000 minutes of sleep (16.5hr) in control flies higher than normal?

Approximately 1000 min of total sleep per day is commonly observed using DAM system. Due to technical limitations, DAM system overestimates amount of sleep, as indicated in Zimmerman et al. (Sleep, 2008, 31(11): 1587-1598).

3. The authors should cite Pirez et al (2019), which shows that s-LNVs signal to mushroom body neurons and this addresses their doubt about LNV-MB connectivity being functional (page 17 in their manuscript).

Indeed. Thank you for the suggestion.

Reviewer #2 (Remarks to the Author):

The premise of this paper is interesting and the approach of RNA-seq validates the initial premise to a large dataset. Essentially, cells with circadian clock mechanisms drive cyclical gene expression of downstream genes, resulting in cell/tissue-specific sets of rhythmic RNA levels that are thought to make a significant contribution to specific rhythmic phenotypes in some aspects of cellular physiology, behavior etc. However, only a small subset of brain neurons have a clock mechanism capable of directly driving endogenous rhythmic gene expression. If cycling mRNA levels is a major aspect of how cells/tissues/organs are regulated in a daily manner, is it possible that clock cells somehow drive cyclical gene expression in non-clock neurons to generate rhythmic behavior? The authors apply this strategy to the mushroom bodies (MBs), a “clockless” brain region critical to sleep, memory, etc, behaviors that are known to exhibit a circadian component. Indeed, they identify 100’s of cycling mRNAs when probing isolated MBs, despite the absence of clock gene expression. Although prior studies have shown this possibility in other tissues, this study generates a very large dataset that can be mined for many uses. In this study the authors focus on a smaller subset of mRNAs that cycle in both LD and DD (117), leading them to test 21 by RNAi in the MBs. 5 of these led to increased sleep in males and females, and for the rest of the paper they focus on Nf1 (Neurofibromin 1) and Pka-C1. Prior work has already shown roles for Nf1 and Pka-C1 in circadian rhythms and sleep in *Drosophila*. The authors provide additional data that Nf1 effects on sleep in the MBs is at least partly due to controlling Pka-C1 activity. Where this manuscript runs into major trouble is with the over-interpretation of their findings. Strip that away and this paper provides; (1) a nice dataset of cyclical gene expression in non-clock containing neurons, and (2) insight into the previously identified MB-wake/sleep regulators Nf1 and Pka-C1, by showing that Nf1 partly functions via Pka-C1 activation.

Thank you for your thorough and constructive assessment of our work.

Below are a few examples of the distance between the data and the claims?:

1. They never show that cyclical RNA expression in the MBs is dependent on functional clocks. While this might seem obvious, especially that some continue in DD, this was not formally shown.

We agree that critical experiment that test this hypothesis was missing. As stated in the answer to Reviewer#1 (major comment 1), we plan to address this issue by conducting Experiment 1.

2. More of a problem is the major claim (see title) that it is the “circadian” cycling of Nf1 and Pka-C1 that is critical to the rhythmic MB-dependent or -driven behaviors previously shown. There are numerous sub-issues within the veracity of such a claim; for example;

a. The authors never show that eliminating the cycling of endogenous Nf1 and/or Pka-C1 mRNA levels in the MBs eliminates a daily behavior. Sure this is very difficult to do but then the claim cannot be made. Overexpressing constitutively activated PKA does not do the trick.

b. Indeed, related to the above, they do not show that Pka-C1 protein levels (and especially) activity undergoes daily rhythms. The same is true for Nf1. This is further compounded by the admitted heterogeneity of the MBs and different functional outcomes depending on spatial concerns. For example, perhaps in the key MB cells driving rhythmic wake-sleep, Pka-C1 is not cycling or shows different phases compared to other regions. Not all cycling mRNAs lead to cycling protein levels. Clearly, the overall cycling of Pka-C1 is consistent with decreased sleep during the day as the authors noted, however....

c. More of a stretch has to be assumed by the authors in building their model in the case of Nf1, as it peaks in the end of the night...e.g., “Since Nf1 mRNA rhythms peak at the end of the night in LD

(Figure 2B), NF1 protein levels likely peak several hours later during the daytime, driving higher wake propensity during the day than at night” (page 11). This begs for measuring Nf1 levels.

Point 2a-c are related problems. To address these issues, we will determine (i) whether and how NF1 and PKA-C1 proteins cycle in the MB, and (ii) whether the cycling of NF1 and PKA-C1 control activity rhythms of MB neurons. Additionally, the reviewer asked (iii) whether eliminating rhythms in NF1 or PKA-C1 levels or activity impairs sleep-wake cycles in a predicted way. As mentioned above, we have already shown that constitutive activation of PKA signaling with UAS-mc* overexpression decreases sleep in a predicted way (Figure 5). This manipulation is not exactly same as eliminating the mRNA rhythms *per se*, but very likely flatten rhythms in downstream signaling. We think that this is a best possible proxy to eliminating mRNA rhythms in the MB, since mechanism generating MB-mRNA rhythms are unknown. Taken together, we plan to collectively address these issues by conducting following experiments.

Experiment 1: Immunostaining of the MB with NF1 and PKA-C1 antibodies, as detailed above.

Experiment 4: Examine calcium levels and cAMP levels of MB neurons at 4 timepoints over 24h in LD, DD and DD in *per0* background in the control (driver only) and in the flies with NF1 knockdown, PKA-C1 knockdown and PKA-C1 mc*overexpression. Calcium levels will be measured using CaLexA-GFP system, and cAMP levels will be measured using the genetically encoded cAMP sensor Epac1-Camps (Yao Z et al. J. Neurophysiol. 2012, 108(2): 684-696).

Importantly, we have already completed part of these experiments. As shown in Figure A displayed at the end of this letter, we measured calcium levels in the MB across 24 h in the control and flies with NF1 knockdown driven with the MB-specific 30Y driver. The calcium levels show daily rhythms with a peak at ZT0. NF1 knockdown significantly reduced the calcium levels at all timepoint, except at ZT12 when calcium levels in the wild-type is at its minimum. These results confirm that excitability of MB neurons is rhythmic and at least partly regulated by the NF1 within the MB. Furthermore, we also found that the calcium levels are significantly downregulated by PKA-C1 knockdown in the MB at least at ZT6 (the only timepoint we have so far tested) (Figure B). Therefore, we are confident that the planned experiment will yield the data, with which we can draw rigorous conclusions.

d. Wake-sleep cycles and the timing of behavior are relatively similar in LD and DD, especially after only 1 day of DD as in the RNA-seq protocol used herein. Yet, the mRNA cycling profiles of *Pka-C1* and *Nf1* are very different (fig. 2B). It is hard to understand how this reasonably explains circadian modulation of sleep drive (process C). This is even more confused by the heterogeneity in different parts of the MBs as either activating wake or promoting sleep. Moreover, especially in LD cycles, activity is bimodal. Yes, flies are more active during the day compared to night, but they have a morning and evening peak of activity. Wherein does a unimodal daily cycle in gene expression (e.g., *Nf1*, *Pka-C1*) drive a bimodal C-process in daily activity?

e. The authors did not eliminate a possible contribution to sleep homeostasis from *Nf1* and *Pka-C1*? Just because its mRNA levels cycle does not mean it can only function in a circadian aspect. Alterations in S-process could appear as alterations in C and vice-versa.

Thank you for your critical analysis of our model. We will carefully revise the model after obtaining all the planned experiments. Since our new results shown in Figure A demonstrates that calcium rhythms in the MB peak during the daytime, it is likely that Process C is the calcium rhythms and/or cAMP rhythms in the MB but there waveform may be different from mRNA expression patterns of NF1 and PKA-C1. (What matters is the rhythms of the critical signaling pathway activity.) Furthermore, to test whether NF1 and PKA-C1 in the MB can be involved in the sleep homeostasis (Process S), we will:

Experiment 5: Examine if NF1 or PKA-C1 knockdown alters rebound sleep after mechanical sleep deprivation.

Essentially, the authors did a very nice study, however they seriously over-interpreted their data to

make quite fundamental claims. I recognize this is difficult terrain but if such claims are to be made they should be justified by the data presented. The evidence that cycling of Nf1 and Pka-C1, at the level of cycling mRNA abundance in the MBs, drives the circadian-dependent bimodal distribution of activity in *Drosophila* is not compelling. More fundamental insights would require, a) showing how clocks drive mRNA cycling in non-clock cells (speculation was put forth on transcription factors and miRNAs, but no evidence was presented), and/or b) show that the actual cycling on Nf1 and Pka-C1 in the MBs is required for the normally observed timing of daily activity in *Drosophila*; i.e., fundamentally tie the rhythmic gene expression to the rhythmic behavior.

Thank you for your positive input. As answered above, we will conduct additional experiments, carefully revise interpretation of the data and tone down the statement wherever relevant. However, to experimentally address . However;

a) showing how clocks drive mRNA cycling in non-clock cells (speculation was put forth on transcription factors and miRNAs, but no evidence was presented)

is out of scope of this study. We provide the plausible hypothesis that transcriptional and miRNA-mediated regulation may be involved in the generation of mRNA rhythms in the MB. In the future, we will generate reporters of MB-cycling genes and mutate relevant TFBSs and miRNA binding sites to test this hypothesis. But these constitute an entirely new project. The point b) will be addressed by Experiment 3, which is explained in the answer to Reviewer #1 (major comments 2).

Reviewer #3 (Remarks to the Author):

The manuscript by Nagoshi and colleagues identifies a significant problem defined by the identification of circadian pacemaker versus non-pacemaker regions of the brain: how is circadian information transmitted from the first to the second? How are specific brain regions circadian-rhythmic if they do not express robust clock proteins in a cell-autonomous fashion? The authors use molecular profiling of purified, genetically-identified neurons to demonstrate daily circadian cycling of ~100 transcripts in neurons that lack expression of canonical clock proteins (PER, TIM etc). These findings make clear that the PER-dependent clock is not required cell-autonomously to drive daily rhythmic gene expression.

The role of NF1 in regulating sleep is well-established so the significance lies centrally on whether there is increased clarification of the role of NF1 in KC (non-clock) cells.

A very recent Science paper by the Reddy lab demonstrates strong circadian control of gene expression in fibroblasts that lack the canonical molecular oscillator (Bmal k/o background). That suggests an ability of eukaryotic cells to generate daily rhythmic patterns without reference to the PER-dependent oscillator. At the least, the authors should acknowledge the possibility that KC cells of the fly brain may possess such abilities, and may therefore not require extrinsic timing information. To address this issue in greater experimental detail, they could knock down Clock or Per in the canonical pacemaker network and confirm continued rhythmic KC gene expression.

Thank you for your constructive comments.

I have several questions about the behavioral studies in this study:

(i) Behavior was tested in conventional fashion (locomotor assays) with 12-32 flies, but seemingly with an N=1; the authors should be clear on whether they tested genotypes once or more than once. From the description offered, it appears to have been once, which is not sufficient – especially with discrepant results between males and females.

Most behavioral experiments were performed more than twice. Only exceptions are the cross that didn't give enough offspring and the initial screening (Figure 2A and S1A-B) (although some of them were tested twice, including NF1 and PKA-C1). As stated in Discussion (line 359-360), we acknowledge that the screening was not exhaustive because it was an exploratory experiment and therefore we focused on the two genes that gave reproducible results in both female and males. In

the revised manuscript, we will more explicitly state our point and indicate the number of experimental replicates. We will also perform repeat experiments wherever missing the replicate.

(ii) The authors created a list of 21 candidates to test based on each of several arbitrary criteria. This is a small list. More remarkable still, each candidate was tested only once (see point above) and apparently only with a single RNAi stock for each. In sum, my criticism is that this 'testing' represents a very narrow technical approach to the experimental question.

We acknowledge the limitations in the screening, and the conclusion was over-stated. In the revised manuscript, we will soften the statement and more clearly explain the limitation of the screening.

(iii) When tested in males and females separately, some genes registered consistently different, but others not so – this was presented without comment or explanation. Perhaps some of these differences represent lack of sufficient replication.

Thank you for pointing this out. We will add this important discussion point in the revised manuscript.

(iv) The results from similar experiments were presented in different formats (violin plots versus histograms): there was no explanation for doing so or what the different formats reveal.

We chose to use violin plot to display initial screening results, as this was the only set of experiments that we evaluate only the total sleep over 24h. Superimposed histograms of 21 candidate genes with control genotypes are simply illegible. We will explain the rationale for the choice of graphs in the Method section in the revised manuscript.

(v) The authors make the point that RNAi stocks tested for 5 of 20 LD-DD candidates affected sleep, whereas, 0 of 12 RNAi sticks for "LD-only" candidates affected sleep. They considered this a significant observation. My question is – on what basis can this be considered significant? But is there any basis to argue? These are extremely small numbers; Was this repeated? Is there a statistic that can be applied to make an argument for significance?

As answered to the points (i)-(iii), we agree with your concern that these are qualitative but not quantitative results. We will clearly soften the statement accordingly in the revised manuscript.

(vi) I could not follow Figure S1 in which the same set of results were said to reveal significant differences (panel A) and no significant differences (panel B). More explanation would be appreciated to understand the nuances presented.

The data sets shown in Figure S1A and S1B were analyzed with statistical tests, but only A showed significant differences in some genotypes. Therefore, we chose to show panel B as a simple graph as it is the negative data. But we can show the Panel B in a violin plot to be consistent with A in the revised manuscript.

The CaLexa results are impressive. However, they are taken at a single arbitrary time point and show a small sub-region of the brain. I recommend displaying a few different time points and showing the whole brain, in addition to the magnified region that describes the KCs.

Your point is well taken. Indeed, as shown in Figure A displayed below, we have already measured CaLexA-GFP levels at 4 timepoints in LD. The new data show that calcium levels cycle in the MB in LD, peaking at ZT0, and NF1 knockdown significantly reduces calcium levels at all timepoints except for ZT12, when calcium rhythms in the control is at its minimum. We did not see any GFP signal outside of the MB as the MB-specific driver was used. However, we will show the snapshot of whole brain and close-up of the KCs in the revised manuscript.

Argument that only a subset of KCs are affected appears not comprehensively-tested

There is no line specific for alpha prime/ beta prime.

In the absence of that, there is no logical basis to exclude a role for alpha prime/ beta prime KCs.

In Discussion (444-446), we have mentioned that the contribution of KCs other than alpha/beta and gamma KCs is not excluded. We will add more thorough discussion in the revised manuscript.

The previous studies of KC regulation of sleep revealed that there is both positive and negative regulation by different subsets. I was not completely persuaded the authors are adequately representing their data to propose a model that is consistent with the previous studies. More clarity on this point is needed.

We agree with your suggestion. We will carefully revise the model and our general conclusion, taking into account the previous studies.

.....
Newly acquired results

Ca²⁺ levels quantified using CaLexA-GFP

Figure A. Calcium levels in all MB lobes show daily rhythms. Fluorescence intensity of CaLexA-GFP driven with 30Y-GAL4 driver were measured every 6h in LD cycles. MB lobes were defined by the co-staining with anti-FasII antibody. Control genotype show daily rhythms in CaLexA-GFP signal, with the peak at ZT0 and the trough at ZT12. NF1 knockdown with *UAS-NF1^{KK}* reduces the CaLexA signal at all timepoints except at ZT12. *p<0.05, ***p<0.001, ****p<0.0001 by 2-way ANOVA with Tukey's multiple-comparisons test. 2 to 3 independent experiments. n=36-52 per group.

CaLexA-GFP levels at ZT6

Figure B. *Pka-C1* knockdown reduces calcium levels in the MB lobes (preliminary results). Calcium levels in all MB lobes were examined by driving CaLexA-GFP with 30Y-GAL4 driver with or without *UAS-Pka-C1* RNAi. Flies were entrained to LD cycles and dissected at ZT6. Single experiment. Control, n=23. PKA-C1 RNAi, n=14. * $p < 0.05$ by Student's t-test.

Reviewers' Comments:

Reviewer #1:

Remarks to the Author:

The authors have responded fairly well to many of my comments and including data from per0 flies is important and an excellent addition at multiple places.

However, I have concerns about the images shown that are being used for quantification.

1. PKA-C1 antibody staining in Figure 3. The images in Figure 3A have a lot of variability in the anti-Fas2 staining, which I believe is being used as a control. If this is correct, then did the authors use the different levels of Fas2 in MBs to correct for differences in PKA-C1 staining? I ask this because there are key differences between levels of PKA-C1 RNA (Figure 2b) and protein (Figure 3a) that are based on the quantification of the data in Figure 3. Specifically, changes in RNA levels seem to precede changes in PKA-C1 protein by about 4 hours in LD. However, in DD PKA-C1 protein levels rise in DD after CT8, even though there are no changes in PKA-C1 RNA levels at in DD after CT4. All of this needs discussing and we need to know if the authors adjusted for different antibody penetration of individual brains to ensure their conclusions are valid. The graph in Figure 3a probably has too many statistical comparisons.

2. The new images in Figure 7 are concerning because of the variability in fluorescence levels within a single brain. For example, the ZT18 show major differences between the left and the right mushroom body calyces in both LD and DD. Furthermore there are large differences in overall fluorescent intensity between time points – for example, ZT6 and ZT12. Does this mean that the cAMP sensor is being rhythmically expressed? Or differentially expressed between left and right MB lobes some of the time?

3. Figure 8 also has very unequal Fas2 staining intensity between timepoints in a single LD cycle, and between left and right MB lobes in the ZT12 image. Does this represent genuine changes in Fas2 levels over time? Or problems with the staining method that would then mean there are problems with interpreting the data?

These issues still need addressing.

Reviewer #2:

Remarks to the Author:

The authors have done a very good job in revising the manuscript to address a key concern. They now show that (at least) PKA-CK1 is under the canonical PER-dependent circadian regulation at the mRNA and protein levels in the MBs despite no local clocks in the MBs. I agree that further details on how this occurs is outside the scope as it could be quite direct (which the authors seem to favor either transcriptional or miRNAs, but this is also due to bioinformatics as a more readily available tool).

They also do a good job in connecting Nf1 to PKA-C1 and activation of MBs, presumably increasing daytime wake (although this was not directly shown as they did RNAi). Certainly there is still some "discrepancies" with prior findings on whether a candidate(Nf1, PKA) in the MBs is wake- or sleep-promoting. It is clear by now that where and/or how much is expressed can alter the balance.

There are a few outstanding issues that I admit were not brought up in my original review but I think are needed for further clarification.

1. The PKA-C1 rhythm shifts in phase between LD and DD1 (this is also the case for the cAMP sensor, fig. 7). Did the authors look at DD1 for the behavior results shown figs 4-6? I know there is no period effect in DD with the MB driver but was there a shift in phase of the RNAi expressing flies relative to control flies? In any case, the authors should make a point about the LD/DD differences in phase for the molecular markers they measured.

2. Related to point #1, it would be good to show the actual locomotor activity profiles and not just the sleep profiles for figs 4-6 (LD, DD1). From the staining of PKA-C1 it seems to coincide with the clock-controlled morning and evening peaks of activity (at least in LD; e.g upswing in late/night--decrease in day, upswing in late evening, decrease during first half of night). In other words, the PKA-C1 daily rhythm appears to be bi-modal (a role in C but not S?). I wonder if an effect on the amplitude/phase of the clock is leading to "indirect" changes in sleep levels observed during the day.

3. Did the authors check activity levels during wake periods to make sure any differences in daytime sleep levels are unrelated to whether the flies are hypo-active?

It might just be the case that whatever is observed in DD is not relevant to what is observed in LD. However, since DD data is shown for the cycling and there is a large shift in phase and it is postulated that the molecular rhythms drive the sleep changes it is important to show the DD behavior.

Reviewer #3:

Remarks to the Author:

The authors have made a substantial effort to consider and address the great majority of criticisms and suggestions presented by the reviewers. The authors have amended their text to align more correctly with the quantitative features of their experiments. The major contribution of this work is to argue for non-autonomous rhythmic gene expression in brain regions that demonstrably important for rhythmic behavior. I feel the additional experiments conducted in a per[0] background go far in arguing that the cycling gene expression in MB neurons is dependent on clock cycling in the pacemaker system, and not downstream of behavior.

Point-by-point reply to reviewers

Reviewer #1 (Remarks to the Author):

*The authors have responded fairly well to many of my comments and including data from *per⁰* flies is important and an excellent addition at multiple places.*

We thank the reviewer for appreciating the improvement in our manuscript. We address all the remaining concerns in this letter and in the revised manuscript. For your perusal, a marked-up copy of the revised manuscript, showing the changes introduced, is uploaded together with a clean copy.

However, I have concerns about the images shown that are being used for quantification.

1. PKA-C1 antibody staining in Figure 3. The images in Figure 3A have a lot of variability in the anti-Fas2 staining, which I believe is being used as a control. If this is correct, then did the authors use the different levels of Fas2 in MBs to correct for differences in PKA-C1 staining? I ask this because there are key differences between levels of PKA-C1 RNA (Figure 2b) and protein (Figure 3a) that are based on the quantification of the data in Figure 3. Specifically, changes in RNA levels seem to precede changes in PKA-C1 protein by about 4 hours in LD. However, in DD PKA-C1 protein levels rise in DD after CT8, even though there are no changes in PKA-C1 RNA levels at in DD after CT4. All of this needs discussing and we need to know if the authors adjusted for different antibody penetration of individual brains to ensure their conclusions are valid. The graph in Figure 3a probably has too many statistical comparisons.

This is an astute observation. We have indeed normalized PKA-C1 antibody staining intensity against Fas2 signal intensity in the data shown in Fig. 3. Same is true for anti-GFP staining in CaLexA-GFP experiments shown in Fig. 8. As shown in the **Figure A** and **B** below, we quantified Fas2 levels to address your question. We found that Fas2 levels had very little differences between timepoints in control flies in LD: ZT12 vs ZT20 showed a statistically significant difference ($p = 0.03$) but no other pairwise comparison showed significant differences (**Fig. A**). In *per⁰* flies in LD, there was no difference in Fas2 across timepoints (**Fig. B**). Importantly, the difference in the Fas2 levels in control genotype (**Fig. A**) is only in one pairwise comparison and very mild compared to the differences in anti-PKA-C1 signals (Fig. 3a). Taken together with the data that PKA-C1 levels are rhythmic in *per⁰* in LD (Fig. 3c) but Fas2 levels do not cycle (**Fig. B**), these results preclude the possibility that a mild fluctuation of Fas2 staining levels artificially created PKA-C1 rhythms.

To avoid confusion, we have also replaced the pictures in Fig. 3a to more representative images.

Figure A. Fas2 levels in the MB lobes during LD.

Anti-Fas2 staining intensity in *w¹¹¹⁸* flies was measured every 4h during LD. The center lines of the box plots indicate median, and the whiskers represent 10 and 90% percentiles. Dots represent the outliers. n=31-39 per group. * $p < 0.05$ by the Kruskal-Wallis ANOVA with Dunn's multiple comparison's test.

Figure B. Fas2 levels in the MB lobes in *per⁰* mutants during LD.

Anti-Fas2 staining intensity in *per⁰* flies was measured every 4h during LD. The center lines of the box plots indicate median, and the whiskers represent 10 and 90% percentiles. Dots represent the outliers. n=30-40 per group. No significant difference was found by the Kruskal-Wallis ANOVA with Dunn's multiple comparison's test.

Specifically, changes in RNA levels seem to precede changes in PKA-C1 protein by about 4 hours in LD. However, in DD PKA-C1 protein levels rise in DD after

CT8, even though there are no changes in PKA-C1 RNA levels at in DD after CT4. All of this needs discussing..

We agreed and added a short discussion on these points (line 201-207).

The graph in Figure 3a probably has too many statistical comparisons.

We admit that Fig. 3a graph looks a little bit messy. However, since this is the results of the statistical test, we think it is appropriate to leave it as it is.

2. The new images in Figure 7 are concerning because of the variability in fluorescence levels within a single brain. For example, the ZT18 show major differences between the left and the right mushroom body calyces in both LD and DD. Furthermore there are large differences in overall fluorescent intensity between time points – for example, ZT6 and ZT12. Does this mean that the cAMP sensor is being rhythmically expressed? Or differentially expressed between left and right MB lobes some of the time?

EPAC1 images shown in Fig.7 were taken from the non-fixed, non-stained brains. Therefore, the fluorescence signal may look sometimes uneven. However, it is just an impression that we perceive from a few samples. Indeed, as shown in below **Figure C**, quantification of CFP/YFP ratios comparing left and right hemispheres did not show any difference both in LD and DD.

Figure C. EPAC1 signal intensity in left and right hemispheres.

The Epac1-camp imaging of relative cAMP levels in MB neurons were performed during LD (Left) and DD (right). CFP/YFP ratios of the signals in the left hemisphere (light blue) and right hemispheres (pink) are shown. The center lines of the box plots indicate median, and the whiskers represent 10 and 90% percentiles. Dots represent the outliers. n=21-34 per group. 2-way ANOVA with Sidak's multiple comparisons test found no difference between left and right hemispheres at any timepoint.

3. Figure 8 also has very unequal Fas2 staining intensity between timepoints in a single LD cycle, and between left and right MB lobes in the ZT12 image. Does this represent genuine changes in Fas2 levels over time? Or problems with the staining method that would then mean there are problems with interpreting the data?

As stated in the answer to the comment 1, Fas2 levels fluctuated very mildly, if at all. We have also replaced the images in Fig. 8a to more representative sets of pictures to avoid confusion.

These issues still need addressing.

Reviewer #2 (Remarks to the Author):

The authors have done a very good job in revising the manuscript to address a key concern. They now show that (at least) PKA-CK1 is under the canonical PER-dependent circadian regulation at the mRNA and protein levels in the MBs despite no local clocks in the MBs. I agree that further details on how this occurs is outside the scope as it could be quite direct (which the authors seem to favor either transcriptional or miRNAs, but this is also due to bioinformatics as a more readily available tool).

They also do a good job in connecting Nf1 to PKA-C1 and activation of MBs, presumably increasing daytime wake (although this was not directly shown as they did RNAi). Certainly there is still some "discrepancies" with prior findings on whether a candidate(Nf1, PKA) in the MBs is wake- or sleep-promoting. It is clear by now that where and/or how much is expressed can alter the balance.

Thank you for the positive comments on our work.

There a few outstanding issues that I admit were not brought up in my original review but I think are needed for further clarification.

We address all the remaining concerns in this letter and in the revised manuscript. For your perusal, a marked-up copy of the revised manuscript, showing the changes introduced, is uploaded together with a clean copy.

1. The PKA-C1 rhythm shifts in phase between LD and DD1 (this is also the case for the cAMP sensor, fig. 7). Did the authors look at DD1 for the behavior results shown figs 4-6? I know there is no period effect in DD with the MB driver but was there a shift in

phase of the RNAi expressing flies relative to control flies? In any case, the authors should make a point about the LD/DD differences in phase for the molecular markers they measured.

2. Related to point #1, it would be good to show the actual locomotor activity profiles and not just the sleep profiles for figs 4-6 (LD, DD1). From the staining of PKA-C1 it seems to coincide with the clock-controlled morning and evening peaks of activity (at least in LD; e.g upswing in late/night--decrease in day, upswing in late evening, decrease during first half of night). In other words, the PKA-C1 daily rhythm appears to be bi-modal (a role in C but not S?). I wonder if an effect on the amplitude/phase of the clock is leading to "indirect" changes in sleep levels observed during the day.

To address both comments 1 and 2, we now display actograms for circadian locomotor behavior in the flies with *Nf1* knockdown, *Pka-C1* knockdown, *Pka-R1(BDK)* expression along with their controls in Supplementary Fig.3a (also manuscript line 241-242). (We did not perform DD experiment with drug-inducible MB-GS and thus these genotypes are not included in this figure.) As you can see, there is no apparent changes in behavioral patterns at the LD to DD transition in any genotype. Therefore, differences in the molecular phases in the MB between LD and DD do not apparently shift behavioral rhythms.

In any case, the authors should make a point about the LD/DD differences in phase for the molecular markers they measured.

This is an excellent suggestion. We added the statements on the LD/DD differences of *Pka-C1* expression (line 201-207) and of cAMP rhythms (line 287-289).

3. Did the authors check activity levels during wake periods to make sure any differences in daytime sleep levels are unrelated to whether the flies are hypo-active?

It might just be the case that whatever is observed in DD is not relevant to what is observed in LD. However, since DD data is shown for the cycling and there is a large shift in phase and it is postulated that the molecular rhythms drive the sleep changes it is important to show the DD behavior.

Following your suggestion, we plotted the mean activity counts during wake period in the flies with *Nf1* or *Pka-C1* knockdown and their controls and display the data in the new Supplementary Fig. 3b. No consistent changes in the activity levels were observed between *Pka-C1* knockdown and two control genotypes. Interestingly, *Nf1* knockdown-flies showed a moderate increase in activity levels compared to two control genotypes. The increase in sleep in caused the *Nf1* knockdown in MB neurons are not due to

hypoactivity. We do not discuss further on the hyperactivity in Nf1 knockdown flies, but it is an interesting subject to pursue in future studies.

Concerning the DD behavior, please see the reply to the comment 1.

.....

Reviewer #3 (Remarks to the Author):

*The authors have made a substantial effort to consider and address the great majority of criticisms and suggestions presented by the reviewers. The authors have amended their text to align more correctly with the quantitative features of their experiments. The major contribution of this work is to argue for non-autonomous rhythmic gene expression in brain regions that demonstrably important for rhythmic behavior. I feel the additional experiments conducted in a *per[0]* background go far in arguing that the cycling gene expression in MB neurons is dependent on clock cycling in the pacemaker system, and not downstream of behavior.*

Reviewers' Comments:

Reviewer #1:

Remarks to the Author:

The authors have responded well to my comments. This is an interesting study - well done!

Reviewer #2:

Remarks to the Author:

The authors have addressed all my comments and have done a good job in improving the manuscript. This a strong addition to circadian control of sleep.

Reviewers' comments:

Reviewer #1 (Remarks to the Author):

The manuscript from Almeida et al examines how circadian rhythms of behavior (sleep) emerge from the activity of a small set of clock neurons that communicate with the rest of the brain to control the behavior of the whole animal – in this case, in the fruitfly.

Almeida et al focus on mushroom body neurons, which have already been implicated by several other groups in control of sleep and activity. Their approach is innovative in that they examine RNA levels in the mushroom bodies and find a set of genes that are rhythmically expressed in light:dark cycles and in constant darkness. This is interesting because the mushroom bodies do not express clock genes, and the authors propose that these rhythms must be being driven by the circadian clock, especially since there are rhythms in constant dark conditions. Then they screen some of these mushroom body genes for altered sleep phenotypes and find many phenotypes. They focus on NF1 and PKA since these proteins fit into a single pathway.

However, I have a number of major problems with the study.

1. The OK107-Gal4 driver line is expressed in the pars intercerebralis in addition to the mushroom body. Images available online seem to also show expression in other regions of the central brain. If this is true, then the conclusions of the study are quite flawed. The authors can address this by showing rhythms of the key genes (Nf1 and Pka-C1) in mushroom body neurons independently of OK107-Gal4, using antibodies or ideally in situ hybridization. This is particularly important since some pars intercerebralis are directly downstream of clock neurons.
2. The authors conclude that the rhythms in RNA levels are controlled by the circadian clock. I agree that this seems likely since some genes show rhythms in DD, which is very interesting. However, the authors do not test if the rhythms depend on clock genes and/or clock neurons, both of which are relatively easily tested. It is quite conceivable that the rhythmic RNA levels are driven by the activity of the fly. Of course, this is controlled by the circadian clock, but this would be a very different conclusion than the one described in the manuscript.
3. The CaLexA experiment in Figure 6 is very interesting, but the FasII staining looks very different with Nf1 RNAi. How did the authors conclude that mushroom body morphology was “unaffected”? Do the authors have the spatial precision to conclude this? Also why change to 30Y as a driver for this CaLexA experiment?
4. Based on the slightly confusing set of drivers used in Figure 6C, where do the authors think that Nf1 is expressed?

Minor comments:

1. For the transcription factor motif study, are these factors expressed in the mushroom body? Are any of them rhythmically expressed? I am not sure if these

data are useful in the manuscript.

2. As a non-sleep expert, are 1000 minutes of sleep (16.5hr) in control flies higher than normal?

3. The authors should cite Pirez et al (2019), which shows that s-LNvs signal to mushroom body neurons and this addresses their doubt about LNV-MB connectivity being functional (page 17 in their manuscript).

Reviewer #2 (Remarks to the Author):

The premise of this paper is interesting and the approach of RNA-seq validates the initial premise to a large dataset. Essentially, cells with circadian clock mechanisms drive cyclical gene expression of downstream genes, resulting in cell/tissue-specific sets of rhythmic RNA levels that are thought to make a significant contribution to specific rhythmic phenotypes in some aspects of cellular physiology, behavior etc. However, only a small subset of brain neurons have a clock mechanism capable of directly driving endogenous rhythmic gene expression. If cycling mRNA levels is a major aspect of how cells/tissues/organs are regulated in a daily manner, is it possible that clock cells somehow drive cyclical gene expression in non-clock neurons to generate rhythmic behavior? The authors apply this strategy to the mushroom bodies (MBs), a “clockless” brain region critical to sleep, memory, etc, behaviors that are known to exhibit a circadian component. Indeed, they identify 100’s of cycling mRNAs when probing isolated MBs, despite the absence of clock gene expression. Although prior studies have shown this possibility in other tissues, this study generates a very large dataset that can be mined for many uses. In this study the authors focus on a smaller subset of mRNAs that cycle in both LD and DD (117), leading them to test 21 by RNAi in the MBs. 5 of these led to increased sleep in males and females, and for the rest of the paper they focus on Nf1 (Neurofibromin 1) and Pka-C1. Prior work has already shown roles for Nf1 and Pka-C1 in circadian rhythms and sleep in *Drosophila*. The authors provide additional data that Nf1 effects on sleep in the MBs is at least partly due to controlling Pka-C1 activity. Where this manuscript runs into major trouble is with the over-interpretation of their findings. Strip that away and this paper provides; (1) a nice dataset of cyclical gene expression in non-clock containing neurons, and (2) insight into the previously identified MB-wake/sleep regulators Nf1 and Pka-C1, by showing that Nf1 partly functions via Pka-C1 activation.

Below are a few examples of the distance between the data and the claims?:

1. They never show that cyclical RNA expression in the MBs is dependent on functional clocks. While this might seem obvious, especially that some continue in DD, this was not formally shown.

2. More of a problem is the major claim (see title) that it is the “circadian” cycling of Nf1 and Pka-C1 that is critical to the rhythmic MB-dependent or -driven behaviors previously shown. There are numerous sub-issues within the veracity of such a claim; for example;

- a. The authors never show that eliminating the cycling of endogenous Nf1 and/or Pka-C1 mRNA levels in the MBs eliminates a daily behavior. Sure this is very difficult to do but then the claim cannot be made. Overexpressing constitutively activated PKA does not do the trick.
- b. Indeed, related to the above, they do not show that Pka-C1 protein levels (and especially) activity undergoes daily rhythms. The same is true for Nf1. This is further compounded by the admitted heterogeneity of the MBs and different functional outcomes depending on spatial concerns. For example, perhaps in the key MB cells driving rhythmic wake-sleep, Pka-C1 is not cycling or shows different phases compared to other regions. Not all cycling mRNAs lead to cycling protein levels. Clearly, the overall cycling of Pka-C1 is consistent with decreased sleep during the day as the authors noted, however....
- c. More of a stretch has to be assumed by the authors in building their model in the case of Nf1, as it peaks in the end of the night...e.g., “Since Nf1 mRNA rhythms peak at the end of the night in LD (Figure 2B), NF1 protein levels likely peak several hours later during the daytime, driving higher wake propensity during the day than at night” (page 11). This begs for measuring Nf1 levels.
- d. Wake-sleep cycles and the timing of behavior are relatively similar in LD and DD, especially after only 1 day of DD as in the RNA-seq protocol used herein. Yet, the mRNA cycling profiles of Pka-C1 and Nf1 are very different (fig. 2B). It is hard to understand how this reasonably explains circadian modulation of sleep drive (process C). This is even more confused by the heterogeneity in different parts of the MBs as either activating wake or promoting sleep. Moreover, especially in LD cycles, activity is bimodal. Yes, flies are more active during the day compared to night, but they have a morning and evening peak of activity. Wherein does a unimodal daily cycle in gene expression (e.g., Nf1, Pka-C1) drive a bimodal C-process in daily activity?
- e. The authors did not eliminate a possible contribution to sleep homeostasis from Nf1 and Pka-C1? Just because its mRNA levels cycle does not mean it can only function in a circadian aspect. Alterations in S-process could appear as alterations in C and vice-versa.

Essentially, the authors did a very nice study, however they seriously over-interpreted their data to make quite fundamental claims. I recognize this is difficult terrain but if such claims are to be made they should be justified by the data presented. The evidence that cycling of Nf1 and Pka-C1, at the level of cycling mRNA abundance in the MBs, drives the circadian-dependent bimodal distribution of activity in *Drosophila* is not compelling. More fundamental insights would require, a) showing how clocks drive mRNA cycling in non-clock cells (speculation was put forth on transcription factors and miRNAs, but no evidence was presented), and/or b) show that the actual cycling on Nf1 and Pka-C1 in the MBs is required for the normally observed timing of daily activity in *Drosophila*; i.e., fundamentally tie the rhythmic gene expression to the rhythmic behavior.

Reviewer #3 (Remarks to the Author):

The manuscript by Nagoshi and colleagues identifies a significant problem defined by the identification of circadian pacemaker versus non-pacemaker regions of the brain: how is circadian information transmitted from the first to the second? How are specific brain regions circadian-rhythmic if they do not express robust clock proteins in a cell-autonomous fashion? The authors use molecular profiling of purified, genetically-identified neurons to demonstrate daily circadian cycling of ~100 transcripts in neurons that lack expression of canonical clock proteins (PER, TIM etc). These findings make clear that the PER-dependent clock is not required cell-autonomously to drive daily rhythmic gene expression.

The role of NF1 in regulating sleep is well-established so the significance lies centrally on whether there is increased clarification of the role of NF1 in KC (non-clock) cells.

A very recent Science paper by the Reddy lab demonstrates strong circadian control of gene expression in fibroblasts that lack the canonical molecular oscillator (Bmal k/o background). That suggests an ability of eukaryotic cells to generate daily rhythmic patterns without reference to the PER-dependent oscillator. At the least, the authors should acknowledge the possibility that KC cells of the fly brain may possess such abilities, and may therefore not require extrinsic timing information. To address this issue in greater experimental detail, they could knock down Clock or Per in the canonical pacemaker network and confirm continued rhythmic KC gene expression.

I have several questions about the behavioral studies in this study:

- (i) Behavior was tested in conventional fashion (locomotor assays) with 12-32 flies, but seemingly with an N=1; the authors should be clear on whether they tested genotypes once or more than once. From the description offered, it appears to have been once, which is not sufficient – especially with discrepant results between males and females.
- (ii) The authors created a list of 21 candidates to test based on each of several arbitrary criteria. This is a small list. More remarkable still, each candidate was tested only once (see point above) and apparently only with a single RNAi stock for each. In sum, my criticism is that this ‘testing’ represents a very narrow technical approach to the experimental question.
- (iii) When tested in males and females separately, some genes registered consistently different, but others not so – this was presented without comment or explanation. Perhaps some of these differences represent lack of sufficient replication.
- (iv) The results from similar experiments were presented in different formats (violin plots versus histograms): there was no explanation for doing so or what the different formats reveal.
- (v) The authors make the point that RNAi stocks tested for 5 of 20 LD-DD candidates affected sleep, whereas, 0 of 12 RNAi sticks for “LD-only” candidates affected sleep. They considered this a significant observation. My question is – on what basis can this be considered significant? But is there any basis to argue? These are extremely small numbers; Was this repeated? Is there a statistic that can be applied to make an argument for significance?
- (vi) I could not follow Figure S1 in which the same set of results were said to reveal

significant differences (panel A) and no significant differences (panel B). More explanation would be appreciated to understand the nuances presented.

The CaLexa results are impressive. However, they are taken at a single arbitrary time point and show a small sub-region of the brain. I recommend displaying a few different time points and showing the whole brain, in addition to the magnified region that describes the KCs.

Argument that only a subset of KCs are affected appears not comprehensively-tested

There is no line specific for alpha prime/ beta prime.

In the absence of that, there is no logical basis to exclude a role for alpha prime/ beta prime KCs.

The previous studies of KC regulation of sleep revealed that there is both positive and negative regulation by different subsets. I was not completely persuaded the authors are adequately representing their data to propose a model that is consistent with the previous studies. More clarity on this point is needed.

Response to Reviewers

Reviewer #1 (Remarks to the Author):

The manuscript from Almeida et al examines how circadian rhythms of behavior (sleep) emerge from the activity of a small set of clock neurons that communicate with the rest of the brain to control the behavior of the whole animal – in this case, in the fruitfly.

Almeida et al focus on mushroom body neurons, which have already been implicated by several other groups in control of sleep and activity. Their approach is innovative in that they examine RNA levels in the mushroom bodies and find a set of genes that are rhythmically expressed in light:dark cycles and in constant darkness. This is interesting because the mushroom bodies do not express clock genes, and the authors propose that these rhythms must be being driven by the circadian clock, especially since there are rhythms in constant dark conditions. Then they screen some of these mushroom body genes for altered sleep phenotypes and find many phenotypes. They focus on NF1 and PKA since these proteins fit into a single pathway.

Thank you for your constructive comments and suggestions. Following your suggestions, we performed several additional experiments and critically edited the manuscript. We believe that these changes significantly improved our manuscript.

However, I have a number of major problems with the study.

1. The OK107-Gal4 driver line is expressed in the pars intercerebralis in addition to the mushroom body. Images available online seem to also show expression in other regions of the central brain. If this is true, then the conclusions of the study are quite flawed. The authors can address this by showing rhythms of the key genes (Nf1 and Pka-C1) in mushroom body neurons independently of OK107-Gal4, using antibodies or ideally in situ hybridization. This is particularly important since some pars intercerebralis are directly downstream of clock neurons.

Your point is well taken. We addressed this issue by conducting immunostaining for PKA-C1 at 6 timepoint under light-dark cycles (LD) and constant darkness (DD), and in DD in *per0* flies. As shown in Fig. 3 of the revised manuscript, we found that PKA-C1 is expressed in rhythmic fashion in the MB lobes and the rhythmicity is abolished in *per0*. We further show that PKA-C1 expression is sometimes detected in the pars intercerebralis (PI) but without rhythms (Supplementary Fig. 2).

We tested several antibodies against NF1, including the one from the lab of Dr. James Walker. However, we could not detect any signal in adult fly brains. Nf1 mRNA levels in the MB is approximately 1/100 of that of Pka-C1 (See Fig. 2B), therefore it is not surprising that NF1 protein was undetectable with immunostaining. However, our new data of PKA-C1 immunostaining unambiguously demonstrate that cell-type specific RNA-seq of OK107-positive neurons was a valid tool to identify genes expressed rhythmically in the MB. It should also be noted that cells in the PI and central brain make up a very small proportion of the OK107-positive cells

compared to the MB intrinsic cells (Kenyon cells, KCs), which count approximately 2000 cells per hemisphere. Therefore, it is very unlikely that RNA-seq counts of cycling genes are entirely from those cycle in non-MB cells. Our data on PKA-C1 immunostaining are in support of this conclusion.

2. The authors conclude that the rhythms in RNA levels are controlled by the circadian clock. I agree that this seems likely since some genes show rhythms in DD, which is very interesting. However, the authors do not test if the rhythms depend on clock genes and/or clock neurons, both of which are relatively easily tested. It is quite conceivable that the rhythmic RNA levels are driven by the activity of the fly. Of course, this is controlled by the circadian clock, but this would be a very different conclusion than the one described in the manuscript.

Thank you for pointing out this important issue. We have performed RNA-seq analysis of MB neurons isolated from *per0* mutants at 6 timepoints in DD and found no gene cycles in this condition (Fig. 1f). This new piece of data indicates that rhythmic gene expression in the MB is driven by circadian clocks that present elsewhere.

3. The CaLexA experiment in Figure 6 is very interesting, but the FasII staining looks very different with Nf1 RNAi. How did the authors conclude that mushroom body morphology was “unaffected”? Do the authors have the spatial precision to conclude this? Also why change to 30Y as a driver for this CaLexA experiment?

We did many replicate experiments and confirmed that morphology of the MB is unaffected by NF1 knockdown. By mistake, we picked a picture that looks a bit deformed, perhaps because the brain was slightly twisted when mounted on the slide glass. We replaced this unrepresentative image. Moreover, we newly performed “around-the-clock” CaLexA imaging experiment at 4 timepoints in LD and DD with and without NF1 or PKA-C1 RNAi, as well as in *per0* background in DD. The results of these experiments demonstrate that CaLexA levels cycle in the MB, driven by circadian clocks, and knockdown of NF1 or PKA-C1 reduces their levels and rhythmicity. The new data are displayed in Fig. 7 and Supplementary Fig. 4.

We used 30Y-GAL4 because it is expressed in all MB lobes and is more practical for crosses. (OK107 is on the 4th chromosome. CaLexA-GFP consists of 3 transgenes. With GAL4 and UAS-RNAi, the lines required for this experiment have 4 or 5 transgenes.)

4. Based on the slightly confusing set of drivers used in Figure 6C, where do the authors think that Nf1 is expressed?

We could not make a definite conclusion from the RNAi experiments using different MB drivers (Fig. 9c of the revised manuscript). However, all the data suggest that the

expression of NF1 in the $\alpha\beta$ and γ lobes is most relevant to the wake-promoting role of NF1. However, this does not exclude the expression and function of NF1 in the $\alpha'\beta'$ lobes.

Minor comments:

1. For the transcription factor motif study, are these factors expressed in the mushroom body? Are any of them rhythmically expressed? I am not sure if these data are useful in the manuscript.

As stated in Discussion (line 368-384), several transcription factors that are predicted to bind MB cycling gene promoters are known to be expressed and function in the MB. We also mention that *cactus*, which encodes the co-factor of one of the enriched transcription factors (Dorsal), is rhythmically expressed in the MB. Therefore, we prefer to display these data (Supplementary Table 2 in the revised manuscript).

2. As a non-sleep expert, are 1000 minutes of sleep (16.5hr) in control flies higher than normal?

Approximately 1000 min of total sleep per day is commonly observed using DAM system. Due to technical limitations, DAM system overestimates amount of sleep, as indicated in Zimmerman et al. (Sleep, 2008, 31(11): 1587-1598).

3. The authors should cite Pirez et al (2019), which shows that s-LNvs signal to mushroom body neurons and this addresses their doubt about LNV-MB connectivity being functional (page 17 in their manuscript).

Indeed. Thank you for the suggestion. This paper is cited in the revised manuscript (line 397-399).

Reviewer #2 (Remarks to the Author):

The premise of this paper is interesting and the approach of RNA-seq validates the initial premise to a large dataset. Essentially, cells with circadian clock mechanisms drive cyclical gene expression of downstream genes, resulting in cell/tissue-specific sets of rhythmic RNA levels that are thought to make a significant contribution to specific rhythmic phenotypes in some aspects of cellular physiology, behavior etc. However, only a small subset of brain neurons have a clock mechanism capable of directly driving endogenous rhythmic gene expression. If cycling mRNA levels is a major aspect of how cells/tissues/organs are regulated in a daily manner, is it possible that clock cells somehow drive cyclical gene expression in non-clock neurons to generate rhythmic behavior? The authors apply this strategy to the

mushroom bodies (MBs), a “clockless” brain region critical to sleep, memory, etc, behaviors that are known to exhibit a circadian component. Indeed, they identify 100’s of cycling mRNAs when probing isolated MBs, despite the absence of clock gene expression. Although prior studies have shown this possibility in other tissues, this study generates a very large dataset that can be mined for many uses. In this study the authors focus on a smaller subset of mRNAs that cycle in both LD and DD (117), leading them to test 21 by RNAi in the MBs. 5 of these led to increased sleep in males and females, and for the rest of the paper they focus on Nf1 (Neurofibromin 1) and Pka-C1. Prior work has already shown roles for Nf1 and Pka-C1 in circadian rhythms and sleep in Drosophila. The authors provide additional data that Nf1 effects on sleep in the MBs is at least partly due to controlling Pka-C1 activity. Where this manuscript runs into major trouble is with the over-interpretation of their findings. Strip that away and this paper provides; (1) a nice dataset of cyclical gene expression in non-clock containing neurons, and (2) insight into the previously identified MB-wake/sleep regulators Nf1 and Pka-C1, by showing that Nf1 partly functions via Pka-C1 activation.

Thank you for your thorough and constructive assessment of our work. Following your advice, we performed additional experiments and critically edited the manuscript, especially to eliminate over-interpretation of the data. We think these changes significantly strengthened our study.

Below are a few examples of the distance between the data and the claims?:

1. They never show that cyclical RNA expression in the MBs is dependent on functional clocks. While this might seem obvious, especially that some continue in DD, this was not formally shown.

We agree that critical experiment that test this hypothesis was missing. As stated in the answer to Reviewer#1 (major comment 2), we conducted RNA-seq analysis of MB neurons in per0 flies in DD. This new set of data demonstrate that gene expression rhythms in the MB are clock-dependent.

2. More of a problem is the major claim (see title) that it is the “circadian” cycling of Nf1 and Pka-C1 that is critical to the rhythmic MB-dependent or -driven behaviors previously shown. There are numerous sub-issues within the veracity of such a claim; for example;

a. The authors never show that eliminating the cycling of endogenous Nf1 and/or Pka-C1 mRNA levels in the MBs eliminates a daily behavior. Sure this is very difficult to do but then the claim cannot be made. Overexpressing constitutively activated PKA does not do the trick.

b. Indeed, related to the above, they do not show that Pka-C1 protein levels (and especially) activity undergoes daily rhythms. The same is true for Nf1. This is further compounded by the admitted heterogeneity of the MBs and different functional outcomes depending on spatial concerns. For example, perhaps in the key MB cells driving rhythmic wake-sleep, Pka-C1 is not cycling or shows different phases compared to other regions. Not all cycling mRNAs lead to cycling protein levels.

Clearly, the overall cycling of Pka-C1 is consistent with decreased sleep during the day as the authors noted, however....

c. More of a stretch has to be assumed by the authors in building their model in the case of Nf1, as it peaks in the end of the night...e.g., “Since Nf1 mRNA rhythms peak at the end of the night in LD (Figure 2B), NF1 protein levels likely peak several hours later during the daytime, driving higher wake propensity during the day than at night” (page 11). This begs for measuring Nf1 levels.

These are fair points. To address point 2a-c, we performed immunostaining of PKA-C1 at 6 timepoints in LD and DD, as well as in *per0* in DD. Please see above the answer to Reviewer 1 (major comment 1). Briefly, we found that PKA-C1 levels cycle in the MB lobes in clock-dependent manner, and its levels peak at ZT8 (Fig. 3). Furthermore, although we were unable to detect NF1 signal in the brain by immunostaining, we demonstrate that calcium levels exhibit circadian rhythms in the MB lobes and these rhythms are impaired when NF1 or PKA-C1 is silenced in the MB. We also show that calcium rhythms in the MB are completely abolished in *per0* flies (Fig. 8). Altogether, these results suggest that the presence of NF1 and PKA-C1 in the MB mediates circadian rhythms of MB activity.

We also restated the phrase “*circadian*” cycling of Nf1 and Pka-C1 that is critical to the rhythmic MB-dependent or -driven behaviors “ throughout the manuscript text and in the title. Additionally, we have removed the data of the overexpression of PKA-C1 *mc** and NF1, since they only indirectly support our conclusion.

d. Wake-sleep cycles and the timing of behavior are relatively similar in LD and DD, especially after only 1 day of DD as in the RNA-seq protocol used herein. Yet, the mRNA cycling profiles of Pka-C1 and Nf1 are very different (fig. 2B). It is hard to understand how this reasonably explains circadian modulation of sleep drive (process C). This is even more confused by the heterogeneity in different parts of the MBs as either activating wake or promoting sleep. Moreover, especially in LD cycles, activity is bimodal. Yes, flies are more active during the day compared to night, but they have a morning and evening peak of activity. Wherein does a unimodal daily cycle in gene expression (e.g., Nf1, Pka-C1) drive a bimodal C-process in daily activity?

Thank you for your critical analysis of our model. Our new results shown in Fig. 7 and 8 demonstrate that cAMP and calcium levels in the MB are higher during the daytime than at night, and their rhythms are driven by circadian clocks and regulated by NF1 and PKA-C1. These findings suggest that the activity of the MB neurons drives Process C, which is mediated by the NF1-cAMP/PKA-C1 pathway. As you correctly pointed out, the waveforms of the *Nf1* and *Pka-C1* mRNA expression are not identical. However, what matters is that the critical signaling pathway activity, i.e. cAMP/PKA-C1 activity just upstream of the calcium increase, controls circadian pattern of intracellular calcium levels in the MB. We would also like to emphasize that, according to the original two-process model by Alexander Borbély posits that the homeostatic sleep drive (sleep debt), Process S, increases during wakefulness and decreases with sleep. Since flies exhibit bimodal activity patterns, we favor the interpretation that fly Process S is bimodal. Whereas process C is the propensity of

the wakefulness, which is independent of the locomotor activity and occurrence sleep/wake. We thus think Process C can be unimodal. We edited the legend of Fig. 9d right panel to clarify the confusion.

By all means, this is simply a model, and we fully acknowledge that there can be alternative explanations on how baseline sleep is regulates in flies.

e. The authors did not eliminate a possible contribution to sleep homeostasis from Nf1 and Pka-C1? Just because its mRNA levels cycle does not mean it can only function in a circadian aspect. Alterations in S-process could appear as alterations in C and vice-versa.

You make a faire point. Indeed, although the two-process model provides a major conceptual framework for understanding sleep, growing evidence suggests that two processes influence each other. To test whether NF1 and PKA-C1 in the MB are involved in the sleep homeostasis, we have performed mechanical sleep deprivation of flies using the Ethoscope system (Geissmann et al. 2017, PLOS Bio). We sleep-deprived the control (*OK107+*) flies and flies in which *Nf1* or *Pka-C1* RNAi was expressed with *OK107-GAL4*.

Figure A. Rebound sleep following the mechanical sleep deprivation. Male (m) and virgin female (f) flies in control and *Nf1* or *Pka-C1* knockdown flies were constantly sleep deprived for 12h during the nighttime. Sleep amount was measured during the first 3 h immediately following the sleep deprivation. 2 independent experiments. control, *OK107/+*. *nf1*, *OK107 > Nf1^{KK}*. *pkac1*, *OK107 > Pka-C1^{KK}*. con, flies without sleep deprivation. SD, flies that underwent sleep deprivation.

As shown in **Figure A**, flies in all genotypes exhibit the trend of homeostatic sleep rebound. However, surprisingly, there was no statistical significance between the control and sleep deprived group, with the exception of *Pka-C1* knockdown in male ($p=0.003$, pairwise Wilcox test). Mechanical sleep deprivation is less consistent than other methods, such as thermogenetic stimulation of wake-promoting neurons (Ref. Vaccaro et al. 2020, Cell), which may have caused the incomplete sleep rebound. Nevertheless, since the overall trend is similar between genotypes, we favor the idea that NF1 and PKA-C1 are not involved in the homeostatic sleep. This is of course a very preliminary result and cannot be displayed on the manuscript. We also believe that pursuing the question whether NF1 and PKA-C1 are involved in the Process S is beyond the scope of the current paper and better suited for a future study.

*Essentially, the authors did a very nice study, however they seriously over-interpreted their data to make quite fundamental claims. I recognize this is difficult terrain but if such claims are to be made they should be justified by the data presented. The evidence that cycling of *Nf1* and *Pka-C1*, at the level of cycling mRNA abundance in the MBs, drives the circadian-dependent bimodal distribution of activity in *Drosophila* is not compelling. More fundamental insights would require, a) showing how clocks drive mRNA cycling in non-clock cells (speculation was put forth on transcription factors and miRNAs, but no evidence was presented), and/or b) show that the actual cycling on *Nf1* and *Pka-C1* in the MBs is required for the normally observed timing of daily activity in *Drosophila*; i.e., fundamentally tie the rhythmic gene expression to the rhythmic behavior.*

Thank you for your positive input. As answered above, we conducted additional experiment, revised the interpretation of the data and tone down the statement wherever relevant. However, to experimentally address;

a) showing how clocks drive mRNA cycling in non-clock cells (speculation was put forth on transcription factors and miRNAs, but no evidence was presented)

is out of scope of this study. We provide the plausible hypothesis that transcriptional and miRNA-mediated regulation may be involved in the generation of mRNA rhythms in the MB. In the future, we will generate reporters of MB-cycling genes and mutate relevant TFBSs and miRNA binding sites to test this hypothesis. But these constitute an entirely new project.

The point b) is addressed in the revised manuscript, as described above.

Reviewer #3 (Remarks to the Author):

The manuscript by Nagoshi and colleagues identifies a significant problem defined by the identification of circadian pacemaker versus non-pacemaker regions of the brain: how is circadian information transmitted from the first to the second? How are specific brain regions circadian-rhythmic if they do not express robust clock proteins in a cell-autonomous fashion? The authors use molecular profiling of purified, genetically-identified neurons to demonstrate daily circadian cycling of ~100 transcripts in neurons that lack expression of canonical clock proteins (PER, TIM etc). These findings make clear that the PER-dependent clock is not required cell-autonomously to drive daily rhythmic gene expression.

The role of NF1 in regulating sleep is well-established so the significance lies centrally on whether there is increased clarification of the role of NF1 in KC (non-clock) cells.

A very recent Science paper by the Reddy lab demonstrates strong circadian control of gene expression in fibroblasts that lack the canonical molecular oscillator (Bmal k/o background). That suggests an ability of eukaryotic cells to generate daily rhythmic patterns without reference to the PER-dependent oscillator. At the least, the authors should acknowledge the possibility that KC cells of the fly brain may possess such abilities, and may therefore not require extrinsic timing information. To address this issue in greater experimental detail, they could knock down Clock or Per in the canonical pacemaker network and confirm continued rhythmic KC gene expression.

Thank you for your constructive comments. We performed several sets of new experiments and critically edited the manuscript to address your concerns. We believe that the revised manuscript is significantly improved following your advice.

At the least, the authors should acknowledge the possibility that KC cells of the fly brain may possess such abilities, and may therefore not require extrinsic timing information. To address this issue in greater experimental detail, they could knock down Clock or Per in the canonical pacemaker network and confirm continued rhythmic KC gene expression.

We newly performed circadian RNA-seq analysis of MB neurons isolated from *per0* flies to address this important point. In Fig.1f of the revised manuscript, we show that no gene cycles in *per0* mutants at in DD. This new piece of evidence supports that rhythmic gene expression in the MB is driven by circadian clocks that present elsewhere.

I have several questions about the behavioral studies in this study:

(i) Behavior was tested in conventional fashion (locomotor assays) with 12-32 flies,

but seemingly with an N=1; the authors should be clear on whether they tested genotypes once or more than once. From the description offered, it appears to have been once, which is not sufficient – especially with discrepant results between males and females.

Most behavioral experiments were performed more than twice. Only exceptions are the cross that didn't give enough offspring and the initial screening (Fig. 2a and Supplementary Fig. 1a, b) (although some of them were tested twice, including NF1 and PKA-C1). As stated in Discussion (line 360-365), we acknowledge that the screening was not exhaustive because it was an exploratory experiment and therefore we focused on the two genes that gave reproducible results in both female and males.

(ii) The authors created a list of 21 candidates to test based on each of several arbitrary criteria. This is a small list. More remarkable still, each candidate was tested only once (see point above) and apparently only with a single RNAi stock for each. In sum, my criticism is that this 'testing' represents a very narrow technical approach to the experimental question.

We acknowledge the limitations in the screening, and the conclusion was overstated. In the revised manuscript, we soften the statement and more clearly explain the limitation of the screening (line 190-192 and 360-365).

(iii) When tested in males and females separately, some genes registered consistently different, but others not so – this was presented without comment or explanation. Perhaps some of these differences represent lack of sufficient replication.

Thank you for pointing this out. We state this important possibility in Discussion (line 364-365) in the revised manuscript.

(iv) The results from similar experiments were presented in different formats (violin plots versus histograms): there was no explanation for doing so or what the different formats reveal.

Agreed. Following your advice, we reformatted all the sleep plots into the box plots.

(v) The authors make the point that RNAi stocks tested for 5 of 20 LD-DD candidates affected sleep, whereas, 0 of 12 RNAi stocks for "LD-only" candidates affected sleep. They considered this a significant observation. My question is – on what basis can this be considered significant? But is there any basis to argue? These are extremely small numbers; Was this repeated? Is there a statistic that can be applied to make an argument for significance?

As answered to the points (i)-(iii), we agree with your concern that these are qualitative but not quantitative results. We removed this statement and simply described the results that no sleep phenotype was found in the experiments when these LD-only cyclers were knocked-down in the MB (line 190-192).

(vi) I could not follow Figure S1 in which the same set of results were said to reveal significant differences (panel A) and no significant differences (panel B). More explanation would be appreciated to understand the nuances presented.

In the previous version of the manuscript, the data sets shown in Figure S1A and S1B were both analyzed with statistical tests, but only A showed significant differences in some genotypes. Therefore, we chose to show panel B as a simple graph as it is the negative data. But we realized that this was confusing indeed. In the revised manuscript, we display both sets of data as box plots (Supplementary Fig. 1a, b).

The CaLexa results are impressive. However, they are taken at a single arbitrary time point and show a small sub-region of the brain. I recommend displaying a few different time points and showing the whole brain, in addition to the magnified region that describes the KCs.

Your point is well taken. Following your suggestion, we performed “around-the-clock” CaLexA imaging experiment at 4 timepoints in LD and DD with and without NF1 or PKA-C1 RNAi, as well as in *per0* background in DD. The results of these experiments demonstrate that CaLexA levels cycle in the MB, driven by circadian clocks, and knockdown of NF1 or PKA-C1 reduces their levels and rhythmicity. The new data are displayed in Fig. 7 and Supplementary Fig. 4.

Argument that only a subset of KCs are affected appears not comprehensively-tested

There is no line specific for alpha prime/ beta prime.

In the absence of that, there is no logical basis to exclude a role for alpha prime/ beta prime KCs.

To further test whether alpha/beta and gamma lobes are important for the wake-promoting role of the NF1-cAMP/PKA-C1 pathway, we newly conducted sleep experiments, in which *Pka-c1* RNAi was driven with some of the MB-drivers. We find that PKA-C1 knockdown in alpha/beta and gamma lobes results in the most consistent daytime sleep increase (Fig. 9b). These results support the idea that alpha/beta and gamma lobes are most relevant for the role of NF1-cAMP/PKA signaling on sleep regulation. However, we agree with your point that GAL4 line specific to the alpha prime/ beta prime lobe was not available, and we could not test the specific requirements of these lobe. In Discussion (line 436-437), we state that the contribution of alpha prime/ beta prime KCs is not excluded.

The previous studies of KC regulation of sleep revealed that there is both positive and negative regulation by different subsets. I was not completely persuaded the authors are adequately representing their data to propose a model that is consistent with the previous studies. More clarity on this point is needed.

We cited literature addressing the positive and negative roles of KC on sleep in Results (line 301-303) and in Discussion (line 425-434). Importantly, previous studies altogether suggest that, although individual KC has positive or negative impact on sleep, activation of several sets of KCs affects sleep in the way that is not simply a sum of the individual effect. We would like to draw your attention previous studies and our study are consistent on this point.

Reviewers' comments

Reviewer #1 (Remarks to the Author):

The authors have responded fairly well to many of my comments and including data from *per0* flies is important and an excellent addition at multiple places.

However, I have concerns about the images shown that are being used for quantification.

1. PKA-C1 antibody staining in Figure 3. The images in Figure 3A have a lot of variability in the anti-Fas2 staining, which I believe is being used as a control. If this is correct, then did the authors use the different levels of Fas2 in MBs to correct for differences in PKA-C1 staining? I ask this because there are key differences between levels of PKA-C1 RNA (Figure 2b) and protein (Figure 3a) that are based on the quantification of the data in Figure 3. Specifically, changes in RNA levels seem to precede changes in PKA-C1 protein by about 4 hours in LD. However, in DD PKA-C1 protein levels rise in DD after CT8, even though there are no changes in PKA-C1 RNA levels at in DD after CT4. All of this needs discussing and we need to know if the authors adjusted for different antibody penetration of individual brains to ensure their conclusions are valid. The graph in Figure 3a probably has too many statistical comparisons.

2. The new images in Figure 7 are concerning because of the variability in fluorescence levels within a single brain. For example, the ZT18 show major differences between the left and the right mushroom body calyces in both LD and DD. Furthermore there are large differences in overall fluorescent intensity between time points – for example, ZT6 and ZT12. Does this mean that the cAMP sensor is being rhythmically expressed? Or differentially expressed between left and right MB lobes some of the time?

3. Figure 8 also has very unequal Fas2 staining intensity between timepoints in a single LD cycle, and between left and right MB lobes in the ZT12 image. Does this represent genuine changes in Fas2 levels over time? Or problems with the staining method that would then mean there are problems with interpreting the data?

These issues still need addressing.

Reviewer #2 (Remarks to the Author):

The authors have done a very good job in revising the manuscript to address a key concern. They now show that (at least) PKA-CK1 is under the canonical PER-dependent circadian regulation at the mRNA and protein levels in the MBs despite no local clocks in the MBs. I agree that further details on how this occurs is outside

the scope as it could be quite direct (which the authors seem to favor either transcriptional or miRNAs, but this is also due to bioinformatics as a more readily available tool).

They also do a good job in connecting Nf1 to PKA-C1 and activation of MBs, presumably increasing daytime wake (although this was not directly shown as they did RNAi). Certainly there is still some "discrepancies" with prior findings on whether a candidate(Nf1, PKA) in the MBs is wake- or sleep-promoting. It is clear by now that where and/or how much is expressed can alter the balance.

There a few outstanding issues that I admit were not brought up in my original review but I think are needed for further clarification.

1. The PKA-C1 rhythm shifts in phase between LD and DD1 (this is also the case for the cAMP sensor, fig. 7). Did the authors look at DD1 for the behavior results shown figs 4-6? I know there is no period effect in DD with the MB driver but was there a shift in phase of the RNAi expressing flies relative to control flies? In any case, the authors should make a point about the LD/DD differences in phase for the molecular markers they measured.

2. Related to point #1, it would be good to show the actual locomotor activity profiles and not just the sleep profiles for figs 4-6 (LD, DD1). From the staining of PKA-C1 it seems to coincide with the clock-controlled morning and evening peaks of activity (at least in LD; e.g upswing in late/night--decrease in day, upswing in late evening, decrease during first half of night). In other words, the PKA-C1 daily rhythm appears to be bi-modal (a role in C but not S?). I wonder if an effect on the amplitude/phase of the clock is leading to "indirect" changes in sleep levels observed during the day.

3. Did the authors check activity levels during wake periods to make sure any differences in daytime sleep levels are unrelated to whether the flies are hypo-active?

It might just be the case that whatever is observed in DD is not relevant to what is observed in LD. However, since DD data is shown for the cycling and there is a large shift in phase and it is postulated that the molecular rhythms drive the sleep changes it is important to show the DD behavior.

Reviewer #3 (Remarks to the Author):

The authors have made a substantial effort to consider and address the great majority of criticisms and suggestions presented by the reviewers. The authors have amended their text to align more correctly with the quantitative features of their experiments. The major contribution of this work is to argue for non-autonomous

rhythmic gene expression in brain regions that demonstrably important for rhythmic behavior. I feel the additional experiments conducted in a *per0* background go far in arguing that the cycling gene expression in MB neurons is dependent on clock cycling in the pacemaker system, and not downstream of behavior.

Point-by-point reply to reviewers

Reviewer #1 (Remarks to the Author):

*The authors have responded fairly well to many of my comments and including data from *per0* flies is important and an excellent addition at multiple places.*

We thank the reviewer for appreciating the improvement in our manuscript. We address all the remaining concerns in this letter and in the revised manuscript. For your perusal, a marked-up copy of the revised manuscript, showing the changes introduced, is uploaded together with a clean copy.

However, I have concerns about the images shown that are being used for quantification.

1. PKA-C1 antibody staining in Figure 3. The images in Figure 3A have a lot of variability in the anti-Fas2 staining, which I believe is being used as a control. If this is correct, then did the authors use the different levels of Fas2 in MBs to correct for differences in PKA-C1 staining? I ask this because there are key differences between levels of PKA-C1 RNA (Figure 2b) and protein (Figure 3a) that are based on the quantification of the data in Figure 3. Specifically, changes in RNA levels seem to precede changes in PKA-C1 protein by about 4 hours in LD. However, in DD PKA-C1 protein levels rise in DD after CT8, even though there are no changes in PKA-C1 RNA levels at in DD after CT4. All of this needs discussing and we need to know if the authors adjusted for different antibody penetration of individual brains to ensure their conclusions are valid. The graph in Figure 3a probably has too many statistical comparisons.

This is an astute observation. We have indeed normalized PKA-C1 antibody staining intensity against Fas2 signal intensity in the data shown in Fig. 3. Same is true for anti-GFP staining in CaLexA-GFP experiments shown in Fig. 8. As shown in the **Figure A** and **B** below, we quantified Fas2 levels to address your question. We found that Fas2 levels had very little differences between timepoints in control flies in LD: ZT12 vs ZT20 showed a statistically significant difference ($p = 0.03$) but no other pairwise comparison showed significant differences (**Fig. A**). In *per⁰* flies in LD, there was no difference in Fas2 across timepoints (**Fig. B**). Importantly, the difference in the Fas2 levels in control genotype (**Fig. A**) is only in one pairwise comparison and very mild compared to the differences in anti-PKA-C1 signals (Fig. 3a). Taken together with the data that PKA-C1 levels are rhythmic in *per⁰* in LD (Fig. 3c) but Fas2 levels do not cycle (**Fig. B**), these results preclude the possibility that a mild fluctuation of Fas2 staining levels artificially created PKA-C1 rhythms.

To avoid confusion, we have also replaced the pictures in Fig. 3a to more representative images.

Figure A. Fas2 levels in the MB lobes during LD.

Anti-Fas2 staining intensity in w^{1118} flies was measured every 4h during LD. The center lines of the box plots indicate median, and the whiskers represent 10 and 90% percentiles. Dots represent the outliers. $n=31-39$ per group. $*p < 0.05$ by the Kruskal-Wallis ANOVA with Dunn's multiple comparison's test.

Figure B. Fas2 levels in the MB lobes in per^0 mutants during LD.

Anti-Fas2 staining intensity in per^0 flies was measured every 4h during LD. The center lines of the box plots indicate median, and the whiskers represent 10 and 90% percentiles. Dots represent the outliers. $n=30-40$ per group. No significant difference was found by the Kruskal-Wallis ANOVA with Dunn's multiple comparison's test.

Specifically, changes in RNA levels seem to precede changes in PKA-C1 protein by about 4 hours in LD. However, in DD PKA-C1 protein levels rise in DD after CT8, even though there are no changes in PKA-C1 RNA levels at in DD after CT4. All of this needs discussing..

We agreed and added a short discussion on these points (line 201-207).

The graph in Figure 3a probably has too many statistical comparisons.

We admit that Fig. 3a graph looks a little bit messy. However, since this is the results of the statistical test, we think it is appropriate to leave it as it is.

2. The new images in Figure 7 are concerning because of the variability in fluorescence levels within a single brain. For example, the ZT18 show major differences between the left and the right mushroom body calyces in both LD and DD. Furthermore there are large differences in overall fluorescent intensity between time points – for example, ZT6 and ZT12. Does this mean that the cAMP sensor is being rhythmically expressed? Or differentially expressed between left and right MB lobes some of the time?

EPAC1 images shown in Fig.7 were taken from the non-fixed, non-stained brains. Therefore, the fluorescence signal may look sometimes uneven. However, it is just an impression that we perceive from a few samples. Indeed, as shown in below **Figure C**, quantification of CFP/YFP ratios comparing left and right hemispheres did not show any difference both in LD and DD.

Figure C. EPAC1 signal intensity in left and right hemispheres.

The Epac1-camp imaging of relative cAMP levels in MB neurons were performed during LD (Left) and DD (right). CFP/YFP ratios of the signals in the left hemisphere (light blue) and right hemispheres (pink) are shown. The center lines of the box plots indicate median, and the whiskers represent 10 and 90% percentiles. Dots represent the outliers. n=21-34 per group. 2-way ANOVA with Sidak's multiple comparisons test found no difference between left and right hemispheres at any timepoint.

3. Figure 8 also has very unequal Fas2 staining intensity between timepoints in a single LD cycle, and between left and right MB lobes in the ZT12 image. Does this

represent genuine changes in Fas2 levels over time? Or problems with the staining method that would then mean there are problems with interpreting the data?

As stated in the answer to the comment 1, Fas2 levels fluctuated very mildly, if at all. We have also replaced the images in Fig. 8a to more representative sets of pictures to avoid confusion.

These issues still need addressing.

Reviewer #2 (Remarks to the Author):

The authors have done a very good job in revising the manuscript to address a key concern. They now show that (at least) PKA-CK1 is under the canonical PER-dependent circadian regulation at the mRNA and protein levels in the MBs despite no local clocks in the MBs. I agree that further details on how this occurs is outside the scope as it could be quite direct (which the authors seem to favor either transcriptional or miRNAs, but this is also due to bioinformatics as a more readily available tool).

They also do a good job in connecting Nf1 to PKA-C1 and activation of MBs, presumably increasing daytime wake (although this was not directly shown as they did RNAi). Certainly there is still some "discrepancies" with prior findings on whether a candidate(Nf1, PKA) in the MBs is wake- or sleep-promoting. It is clear by now that where and/or how much is expressed can alter the balance.

Thank you for the positive comments on our work.

There a few outstanding issues that I admit were not brought up in my original review but I think are needed for further clarification.

We address all the remaining concerns in this letter and in the revised manuscript. For your perusal, a marked-up copy of the revised manuscript, showing the changes introduced, is uploaded together with a clean copy.

1. The PKA-C1 rhythm shifts in phase between LD and DD1 (this is also the case for the cAMP sensor, fig. 7). Did the authors look at DD1 for the behavior results shown figs 4-6? I know there is no period effect in DD with the MB driver but was there a shift in phase of the RNAi expressing flies relative to control flies? In any case, the authors should make a point about the LD/DD differences in phase for the molecular markers they measured.

2. Related to point #1, it would be good to show the actual locomotor activity profiles and not just the sleep profiles for figs 4-6 (LD, DD1). From the staining of PKA-C1 it

seems to coincide with the clock-controlled morning and evening peaks of activity (at least in LD; e.g upswing in late/night--decrease in day, upswing in late evening, decrease during first half of night). In other words, the PKA-C1 daily rhythm appears to be bi-modal (a role in C but not S?). I wonder if an effect on the amplitude/phase of the clock is leading to "indirect" changes in sleep levels observed during the day.

To address both comments 1 and 2, we now display actograms for circadian locomotor behavior in the flies with Nf1 knockdown, *Pka-C1* knockdown, *Pka-R1(BDK)* expression along with their controls in Supplementary Fig.3a (also manuscript line 241-242). (We did not perform DD experiment with drug-inducible MB-GS and thus these genotypes are not included in this figure.) As you can see, there is no apparent changes in behavioral patterns at the LD to DD transition in any genotype. Therefore, differences in the molecular phases in the MB between LD and DD do not apparently shift behavioral rhythms.

In any case, the authors should make a point about the LD/DD differences in phase for the molecular markers they measured.

This is an excellent suggestion. We added the statements on the LD/DD differences of *Pka-C1* expression (line 201-207) and of cAMP rhythms (line 287-289).

3. Did the authors check activity levels during wake periods to make sure any differences in daytime sleep levels are unrelated to whether the flies are hypo-active?

It might just be the case that whatever is observed in DD is not relevant to what is observed in LD. However, since DD data is shown for the cycling and there is a large shift in phase and it is postulated that the molecular rhythms drive the sleep changes it is important to show the DD behavior.

Following your suggestion, we plotted the mean activity counts during wake period in the flies with *Nf1* or *Pka-C1* knockdown and their controls and display the data in the new Supplementary Fig. 3b. No consistent changes in the activity levels were observed between *Pka-C1* knockdown and two control genotypes. Interestingly, *Nf1* knockdown-flies showed a moderate increase in activity levels compared to two control genotypes. The increase in sleep in caused the *Nf1* knockdown in MB neurons are not due to hypoactivity. We do not discuss further on the hyperactivity in *Nf1* knockdown flies, but it is an interesting subject to pursue in future studies.

Concerning the DD behavior, please see the reply to the comment 1.

REVIEWERS' COMMENTS

Reviewer #1 (Remarks to the Author):

The authors have responded well to my comments. This is an interesting study - well done!

Reviewer #2 (Remarks to the Author):

The authors have addressed all my comments and have done a good job in improving the manuscript. This a strong addition to circadian control of sleep.